# Exploring the coupling coordination relationship and obstacle factors of rural revitalization, new-type urbanization, and digital economy in China

Yajun Ma[1], Zhengyong Yu[2]\*, Wei Liu[3]\*, Qiang Ren[4]

1 College of Tourism, Xinyang Normal University, Xinyang, China, 2 School of Business and Tourism Management, Yunnan University, Kunming, China, 3 School of Environment, Tsinghua University, Beijing, China, 4 Business School, Sichuan University, Chengdu, China

\* yuzhengyong@stu.ynu.edu.cn (ZY); xueshu2019lw@163.com (WL)

**Data Availability Statement:** All relevant data are within the manuscript.

**Funding:** The authors gratefully acknowledge the financial support funded by the Humanities and

## Abstract

The digital economy injects new vitality into rural revitalization and new-type urbanization to achieve rural industrial transformation, while the countryside and the city provide the soil for the development of the digital economy. This research establishes the rural revitalization (RR), new-type urbanization (NU), and digital economy (DE) system and uses the coupled coordination degree (CCD) model and obstacle degree (OD) model to study the spatiotemporal evolution characteristics and obstacle factors of the composite system in China from 2011 to 2021. The result showed that: (1) the comprehensive development level of the composite systems is on an upward trend year by year, but still shows a low-quality state; (2) the CCD of China's provinces shows a spatial evolution pattern of high in the east and low in the west; (3) The obstacle factors of the RR, NU, DE subsystem are mainly involved the number of rural doctors and health workers, local financial income per capita and science and technology expenditure, and the digital finance coverage breadth index. These results suggested that Strengthening the synergy between China's urban-rural integrated development and digital construction in the future hinges upon providing valuable decision-making insights to facilitate the pursuit of regionally differentiated development and the achievement of sustainable development goals.

## 1. Introduction

Inequality is a global concern of the United Nations Sustainable Development Goals, and the digital economy (DE) is helping to reduce a country's income gap on a global scale [1]. With the arrival of the digital economy, the number of global Internet users has reached 5.373 billion. The digital economy breaks the limitations of traditional modes, weakens the asymmetry of information access, and becomes an efficient driving force for economic efficiency and quality improvement [2]. The United Nations introduced the Sustainable Development Goals (SDGs), which, in order to accelerate progress and achievements, specifically emphasize the

Social Science Fund of the Ministry of Education (Grant No. 22YJA630060, 22YJC790039) and supported by Yunnan Provincial Department of Education Science Research Fund (Grant No. 2024Y00504, 2023Y0363). This research is also supported by grants from Yunnan University's 15th Key Project of Scientific Research and Innovation (Grant No. KC-23235237) The funders had no role in study design, data collection and analysis, decision to publish, or preparation of the manuscript.

**Competing interests:** The authors have declared that no competing interests exist

importance of the countryside and promote rural revitalization (RR) as a critical strategic objective [3]. The ultimate goal of the rural revitalization strategy is common prosperity, which solves the problem of unbalanced development between urban and rural areas. New-type urbanization (NU), as an exogenous power of the countryside, promotes the flow of talents, science and technology, capital, information, and other factors to the countryside, stimulates the endogenous power of the countryside, and realizes the revitalization of the countryside. RR is the foundation and premise of new urbanization, which brings new space carriers and demand power for the development of DE. At the same time, DE provides new kinetic energy, such as information and resources for the development of rural revitalization. In the process of promoting rural revitalization, accelerating new urbanization and developing digital economy, exploring the coupled and coordinated relationship, spatial and temporal evolution pattern, and the obstacle factors of the coordinated development in the RR-NU-DE system will help to accelerate the realization of a series of goals, such as China's economic transformation, digitalization, urban-rural integration, and rural revitalization, and at the same time refer to the digital economy and urban-rural integration in the world. From a global perspective, many countries are currently facing unbalanced development between urban and rural areas, aging populations, technological backwardness, and other dilemmas, and exploring the coupling relationship between rural revitalization, new urbanization, and the digital economy, as well as the main obstacles, can effectively reduce the gap between urban and rural areas, improve the overall competitiveness of the country, and promote the flow of resources between regions, thereby promoting the balanced development of all countries globally and realizing the optimization of the allocation of resources globally.

RR is the ultimate goal of rural development. The International Fund for Agricultural Development has improved rural economic and environmental sustainability in a transformational direction, the European Union has made sustainable agriculture and villages one of its development goals, and the Organization for Economic Cooperation and Development has made sustainable agricultural growth a strategic objective [4–6]. Scholars have already researched the value, influencing factors, and practical path of RR. The separation of urban and rural areas is an inevitable product of historical development. RR is an essential means of resolving the urban-rural dichotomy coordination development of urban and rural areas in urban-rural separation [7], as well as an effective way to solve the problem of underutilization of land resources due to the loss of young talents [8]. The rapid development of the rural economy and the promotion of rural revitalization contribute to a good external economic image [9]. RR is affected by various factors in the development process, such as the natural and social environment, infrastructure, energy use, and agricultural productivity [10]. Addressing the challenges of urban-rural polarization and rural decline, we should take targeted measures to promote revitalization within broader rural development initiatives effectively. We should make agriculture a priority industry, formulate scientific plans, build a sustainable agro-eco-system, and develop infrastructure to promote rural revitalization.

Furthermore, rural development can be fostered by implementing targeted subsidy policies and initiatives to enhance the human capital of rural populations [11]. Over time, urbanization has shown an irreversible trend, with a series of problems such as land use and environmental pollution, and various countries have put forward strategies such as "sustainable cities," "resilient cities," and "new-type urbanization" to cope with this problem [12–15]. Urban-rural transformation results from the interaction between rural development (internal drivers) and urban development (external drivers) [16]. Currently, some scholars mainly focus on the definition of connotation, influencing factors, level measurement, and role in urbanization. Urbanization is the process of transferring agricultural households from the countryside to the city and is accompanied by changes in production and lifestyles [17]. Changes in the

demographic structure are the main feature of urbanization—factors such as population movements, quality, and infrastructure influence urban development. With regard to the level of urbanization, there has been a shift from single variables, such as the ratio of the urban population to the total population, to composite variables, such as economic, demographic, healthcare, and industrial education [18]. Common methods for measuring the level of urbanization include life cycle assessment, cost-benefit analysis, and regional planning models [19–21]. NU promotes the flow of resource factors and the interaction of internal and external push and pull forces in urban and rural areas to promote the diversification of food production in suburban areas [16, 22].

The digital economy is based on information technology as the core, widely cited as the driving force to promote social and economic development. The digital economy is the representative of advanced productivity. Scholars in the DE research mainly focus on measurement, impact effects, and other aspects. The measurement of the DE is currently focused on two frameworks, direct and indirect measurement, with direct measurement referring to the measurement of the scale of value added corresponding to the DE by directly adopting the connotation and statistical scope of the DE [23]. Indirect measurement refers to the use of multiple indicators and dimensions to construct an analytical framework by which to measure the DE of a given economy [24]. The modeling concludes that digital finance is vital in strategic emerging companies. The digital economy has penetrated all aspects of life and various industries. It can effectively promote technological innovation, adjust industrial structure, improve total factor productivity, increase employment opportunities, and thus promote social governance. In the industrial transformation process, the digital economy can optimize industrial structure and human structure, improve innovation ability, and promote the integration and development of manufacturing, financial industry, and service industries. At the same time, the digital economy empowers the high-quality development of the economy through technological innovation, innovation activity, Internet popularization, and other paths [25].

The DE has permeated all aspects of life, and it can effectively promote technological innovation and adjust the industrial structure while driving up the level of total factor productivity and increasing employment opportunities, thereby promoting social governance. Scholars have made a lot of research results on the coupling and coordination of DE and NU. The development of NU has encountered many difficulties. The use of digital economy technology can help promote the standardization and intelligent construction of NU, such as building a platform for urban resources, promoting the flow of resource elements, and improving the quality of the economy while promoting the development of urbanization [26]. Distinct from traditional urbanization, the digital economy brings innovation-driven and investment-driven new urbanization, empowering innovative mechanisms and accelerating local urbanization [27]. The digital economy, under powerful information network technology, breaks down spatial barriers, promotes economic development at the macro level, transforms traditional industries, and raises the income of farmers, thus improving people's living standards and narrowing the gap between urban and rural areas, and strengthens enterprise cooperation at the micro level, driving a profound butterfly change in the development and construction of urban and rural areas [28]. NU and DE are highly consistent regarding development objectives, and new-type urbanization provides spatial carriers, specific application scenarios, and potential demand for developing a digital economy [29].

The relationship between RR and the DE is also one of the hot topics of research and discussion. The construction of smart villages through communication technology improves the quality of survival of farming households, promotes employment, and raises wages for farming households through the strengthening of digital facilities, such as broadband construction, reduces the gap between urban and rural areas, improves the competitiveness of villages and

promotes rural revitalization [30]. The network effect of the DE has led to the development of the e-commerce industry, injected new vitality into the rural economy, and brought digital economic thinking to the countryside, breaking down traditional rural interpersonal relationships, reshaping rural grass-roots governance, and prompting the digitalization of rural revitalization. Digital agriculture and digital finance are all important parts of the digital economy. Among them, DE is critical to integrating the digital economy and rural revitalization. The DE can promote rural revitalization by improving productivity in the agricultural industry, optimizing production relations, and improving the marketability of products [31]. Scholars have demonstrated from various perspectives that the digital economy can promote rural revitalization with strong correlations, but there is still a need to explore high-level coupled and coordinated hierarchical pathways [32]. However, some scholars have pointed out that with the rapid development of the DE, a "digital divide" has emerged in urban and rural development, with the rich having access to more information and policy dividends to raise their incomes. The gap between them and those with less Internet access has widened, thus hindering rural development and exacerbating the impoverishment of the countryside [33, 34]. With the advancement of digital technology and the process of rural construction, digital villages and rural digitization have gradually become essential connotations of rural revitalization, and the orderly development of rural revitalization has provided a solid material foundation and hardware facilities for the development of the digital economy.

Regarding the relationship among RR, NU, and DE, scholars have mostly focused on exploring the relationship between a single system or two or two systems. Whether the relationship between RR and NU is mutually reinforcing or mutually inhibiting has been widely discussed. The most representative dual structure theory points out that agriculture and industry are the basic economic sectors and that cities can drive the development of the countryside and redirect surplus labor from the countryside to the industrial sector [35]. The integration of the countryside and the city, which are the basic forms of society, promotes economic development, raises the incomes of agricultural households by accelerating the process of urban-rural integration, shifts agricultural households from agricultural to non-agricultural activities, and thus reduces the expenditures of agricultural households and the incidence of poverty, thereby contributing to the development of the countryside [36]. The fundamental key to the RR strategy is to solve the "three rural issues," the core of which is urbanization. NU promotes economic development, and rural revitalization eliminates social risks, so NU and RR should work in tandem.

Moreover, the synergistic development of these two approaches can enhance economic circulation, fostering a gradual equilibrium between rural and urban areas. NU facilitates the flow of resources between cities and villages, promoting both rural development and RR. Concurrently, NU can effectively address urban challenges and mitigate issues encountered during its development process. However, some scholars believe that the development of NU has led to the underdevelopment of rural resources, the inequality of urban and rural environments, and the emergence of the phenomenon of hollowing out of the countryside, with the development gradually lagging [37, 38]. Scholars' debates and analyses on the relationship between the two have become more and more in-depth, along with the advancement of RR and the NU process, and there is a certain degree of variability in their research perspectives with different case regions.

However, existing research has not yet included the DE, NU, and RR in the overall research framework, and exploring the coupling and coordination level, spatial and temporal evolution, and influencing factors of the three is conducive to a better understanding of the inherent logical relationship between the three, which is conducive to promoting rural revitalization, common prosperity, and modernization goals. Given this, this research measures the coupling and

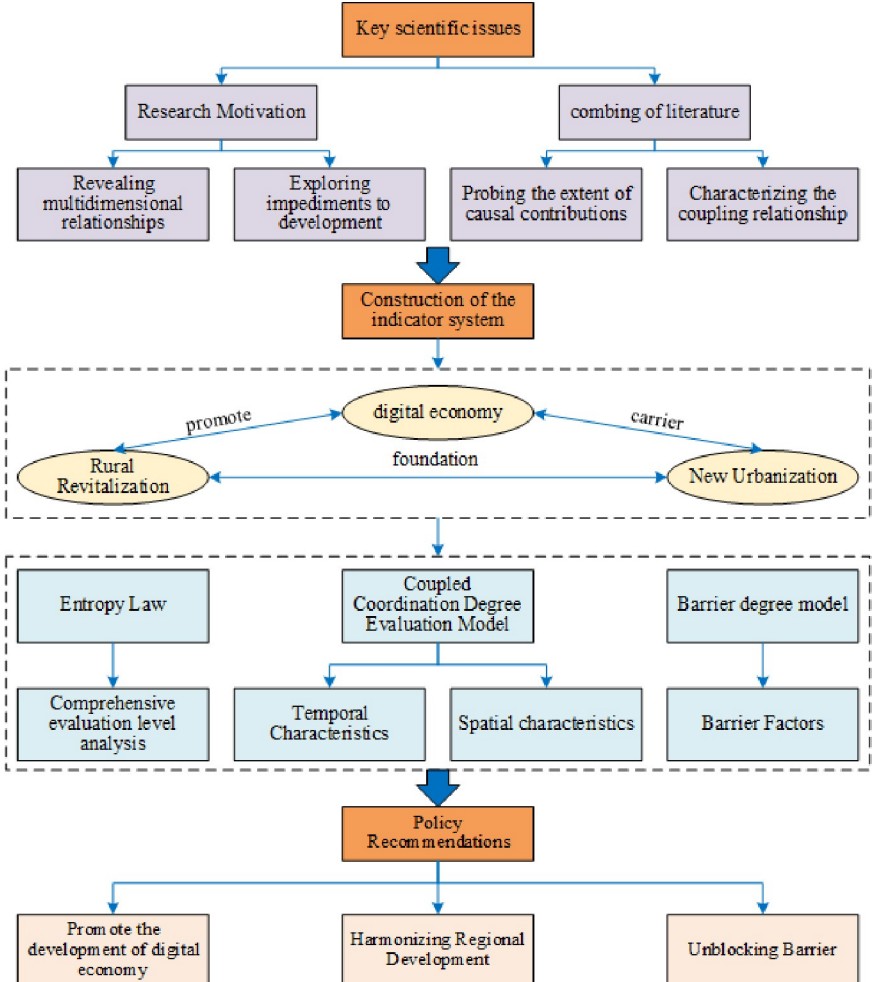

**Fig 1. The logical framework of China's rural revitalization, new-type urbanization, and digital economy composite system.**

coordination level of DE, NU, and RR based on the 2011–2021 panel data of 31 provinces across China. It measures the influencing factors through the obstacle level, intending to provide scientific theoretical guidance for the synergistic development of DE, NU, and RR in China. The logical framework of China's RR-NU-DE composite system is shown in Fig 1.

## 2. The establishment of the indicator system and evaluation model for the RR-NU-DE complex system

### 2.1 Construction of the indicator system

This research focuses on the role mechanism of rural revitalization, new-type urbanization, and digital economy coupling and coordinated development. It also constructs the RR-NU-DE indicator system in 31 provinces. The construction of the assessment system indicators follows the principles of scientificity, comprehensiveness, accessibility, etc. The screening of DE assessment indicators is based on compliance with the laws of DE development and the accessibility of digital economy indicator data; the screening of NU assessment indicators is based on the structure of the composition of the NU system; and the rural revitalization assessment

indicator system can accurately measure the twenty-word guidelines of the rural revitalization strategy.

After reviewing the existing literature, based on the existing indicator system, and in conjunction with the 20-word guideline of the rural revitalization strategy, we have identified 15 secondary indicators based on the measurement dimensions of five specific objectives, namely, "prosperous industry, ecological livability, civilized countryside, effective governance, and common prosperity." The connotations are as follows: industrial prosperity is the prerequisite for increasing rural productivity and is the foundation and critical support for rural revitalization; ecological livability is an essential criterion for measuring the sustainable development of the countryside and is the environmental support for rural revitalization; a civilized countryside is the key to promoting the ideological construction of farmers and raising their awareness; effective governance is an important guarantee for the modernization of rural governance; and common prosperity is the ultimate goal of rural revitalization.

According to the relevant National New-type Urbanization Plan (2014–2020) plan, the new-type urbanization evaluation index system is constructed from 17 secondary indicators in 5 dimensions combined with existing research. Population dimension: In the process of new-type urbanization, the population gradually gathers in cities, and the composition of employment is adjusted [39]. The population dimension reflects the level of population urbanization; the Economic dimension: economic growth in cities and villages and industrial restructuring in the new-type urbanization process [40]. The level of economic development is the fundamental driving force of urbanization, which reflects the level of economic urbanization through the economic dimension. Spatial dimension: spatial urbanization mainly measures the coverage of towns and cities and the rate of intensive land use [41, 42]. The spatial dimension focuses on the spatial logic of new urbanization and rural revitalization and emphasizes the integration and optimization of urban and rural space to form a new urbanization development path that transcends the traditional urban-rural boundaries. Social dimension: this encompasses the transformations in infrastructure and shifts in residents' lifestyles accompanying the new-type urbanization process [43]. It is an essential requirement of the development of new urbanization. Ecological dimension: changes in the living environment of residents, the relationship between urban construction and resources, and the environment in the process of new-type urbanization [44]. Measuring the sustainable development of new urbanization and the urban-rural integration and coordination dimension is necessary. This indicator reflects the degree of coordination between urban and rural development.

With reference to the Global Digital Economy White Paper and existing studies, the digital economy evaluation index system is constructed from 16 secondary indicators in four dimensions: digital research and innovation, digital inclusive finance, digital industrialization, and digital infrastructure. Digital infrastructure: digital infrastructure is the carrier of the digital economy, measuring the ability of the digital economy in terms of computing security services [45]; digital industrialization dimension: the digital industry is the driving force for the development of the digital economy, providing related technical services and other economic activities for the development of digital economy [46]. From the macro level, it reflects the development ability of the ICT industry in the digital economy; digital scientific research and innovation dimension: digital scientific research and innovation measure the degree of intelligent technology and innovation level of the digital economy [47]; digital financial inclusion dimension: the degree of digital financial inclusion can be judged by the degree of urban and rural areas. The degree of digital inclusive finance can judge the income gap between urban and rural digital economies [48]. The weights of the indicators are calculated by the entropy method (Table 1).

Pearson correlation research is a statistical analysis method proposed by British mathematician Carl Pearson. The Pearson correlation coefficient is between [–1,1] to quantify the correlation between variables. The stronger the linear relationship between the two variables, the more the correlation coefficient tends to -1 or 1. Figs 2–4 are obtained by Pearson correlation analysis.

From Figs 2 and 4, it can be seen that the correlation coefficients of each index in the rural revitalization system and the new urbanization system are concentrated between [0,1], with obvious correlation. In Fig 3, the correlation coefficients of each index in the digital economy system are all above 0, and all indexes are correlated.

**Table 1. Evaluation index system of the RR-NU-DE composite system.**

|  | Dimension | Index | Unit | attributes | Weights |
|---|---|---|---|---|---|
| New-type urbanization (X) | Economy | Per capita disposable income of urban residents(x1) | yuan | + | 0.041 |
|  |  | Per capita fixed-asset investment (x2) | yuan | + | 0.028 |
|  |  | The ratio of the output value of the secondary industry to the tertiary industry (x3) | % | + | 0.059 |
|  |  | Share of output value of secondary and tertiary industries (x4) | % | + | 0.010 |
|  |  | Per capita retail sales of social consumer goods (x5) | yuan | + | 0.039 |
|  |  | Per capita GDP(x6) | yuan | + | 0.043 |
|  |  | Per capita local fiscal revenue (x7) | yuan | + | 0.071 |
|  | Space | Per capita mileage of highway routes (x8) | kilometres | + | 0.073 |
|  |  | Road network density (x9) | Kilometres/km2 | + | 0.040 |
|  |  | Per capita urban road area (x10) | $m^2$ | + | 0.017 |
|  |  | Per capita built-up area (x11) | $m^2$ | + | 0.064 |
|  | Society | Per capita local fiscal expenditure on science and technology (x12) | yuan | + | 0.110 |
|  |  | Public transport vehicles per 10,000 people (x13) | unit | + | 0.017 |
|  |  | Per capita local fiscal expenditure on education (x14) | yuan | + | 0.044 |
|  |  | Number of health technicians per 10,000 people (x15) | people | + | 0.022 |
|  |  | Public toilets per 10,000 people (x16) | unit | + | 0.022 |
|  |  | Internet penetration rate (x17) | % | + | 0.021 |
|  |  | Urban gas penetration rate (x18) | % | + | 0.003 |
|  |  | Urban water penetration rate (x19) | % | + | 0.001 |
|  |  | Share of social security and employment expenditure in fiscal expenditure (x20) | % | + | 0.019 |
|  | Population | Urbanization rate (x21) | % | + | 0.013 |
|  |  | Urban population density (x22) | people/km2 | + | 0.022 |
|  |  | Share of employees in secondary and tertiary industries (x23) | % | + | 0.017 |
|  |  | The registered unemployment rate of the urban population (x24) | % | - | 0.011 |
|  |  | Students enrolled in higher education per 100,000 population (x25) | people | + | 0.024 |
|  | Ecology | Green coverage rate of urban built-up area (x26) | % | + | 0.003 |
|  |  | Per capita public green space area (x27) | $m^2$ | + | 0.014 |
|  |  | Treatment rate of domestic sewage (x28) | % | + | 0.003 |
|  |  | Per capita sulfur dioxide emissions (x29) | t/people | - | 0.084 |
|  |  | Harmless treatment ratio for house refuse (x30) | % | + | 0.006 |
|  | Urban and rural coordination degree | Comparison of urban and rural residents consumption level (x31) | % | + | 0.026 |
|  |  | Comparison of urban and rural residents income level (x32) | % | + | 0.026 |
|  |  | Engel coefficient ratio of urban and rural households (X33) | % | + | 0.009 |

(*Continued*)

**Table 1.** (Continued)

| | Dimension | Index | Unit | attributes | Weights |
|---|---|---|---|---|---|
| Rural revitalization (y) | Industrial prosperity | Per capita primary industry output value (y1) | billions | + | 0.066 |
| | | Grain output (y2) | hectares | + | 0.037 |
| | | Total power of agricultural machinery (y3) | kilowatt | + | 0.037 |
| | | Effective irrigation area (y4) | thousand hectares | + | 0.049 |
| | Ecological livable | Percentage of forest cover (y5) | % | + | 0.097 |
| | | Solid waste treatment rate (y6) | % | + | 0.014 |
| | | Number of public toilets (y7) | unit | + | 0.011 |
| | | Number of rural doctors and health workers (y8) | 万人 | + | 0.162 |
| | | Comprehensive production capacity of water supply (y9) | cubic metre /day | + | 0.024 |
| | Rural civilization | Public education expenditures of local government (y10) | billions | + | 0.027 |
| | | The ratio of the number of persons aged 15 and over to the number of persons aged 15 and over who are illiterate (y11) | % | + | 0.209 |
| | | Comprehensive population coverage of TV programs (y12) | % | + | 0.013 |
| | Effective governance | The ratio of disposable income per rural resident to disposable income per urban resident (y13) | % | + | 0.051 |
| | | The ratio of per capita consumption expenditure of rural residents to per capita consumption expenditure of urban residents (y14) | % | + | 0.030 |
| | | Rural population (y15) | people | + | 0.076 |
| | Live in plenty | Per capita disposable income of rural residents (y16) | yuan | + | 0.038 |
| | | Rural consumption expenditure on food (y17) | yuan | + | 0.028 |
| | | The number of rural individual employment (y18) | people | + | 0.031 |
| Digital economy (z) | Digital scientific research innovation | The number of patent application authorizations (z1) | Pieces | + | 0.085 |
| | | Total technical contract turnover (z2) | yuan | + | 0.123 |
| | | The number of R & D projects of industrial enterprises above the designated size (z3) | item | + | 0.083 |
| | | R & D expenditure of industrial enterprises above designated size (z4) | yuan | + | 0.070 |
| | Digital inclusive finance | Digital financial digitization index (z5) | / | + | 0.014 |
| | | Digital finance usage depth index (z6) | / | + | 0.014 |
| | | Digital financial coverage breadth index (z7) | / | + | 0.225 |
| | Digital industrialization | Software industry revenue (z8) | yuan | + | 0.108 |
| | | Telecommunication traffic (z9) | billions | + | 0.070 |
| | Digital infrastructure | The number of Internet Domain Names (z10) | unit | + | 0.081 |
| | | The number of Internet broadband access ports (z11) | unit | + | 0.038 |
| | | Mobile subscription (z12) | % | + | 0.014 |
| | | The number of mobile phone base stations (z13) | unit | + | 0.037 |
| | | Cable length (z14) | kilometres | + | 0.038 |

Note: "+" and "-" represent the positive and negative indicators.

## 2.2 Evaluation index and model construction of RR-NU-DE composite system

**2.2.1 Evaluation model of comprehensive development level.** The comprehensive development level of the system is explored by constructing a comprehensive evaluation index model for the RR, NU, and DE systems. The three systems of RR, NU, and DE are independent and interact with each other, and the "total contribution" of each subsystem indicator to the three systems can be obtained through the level of comprehensive development, reflecting the

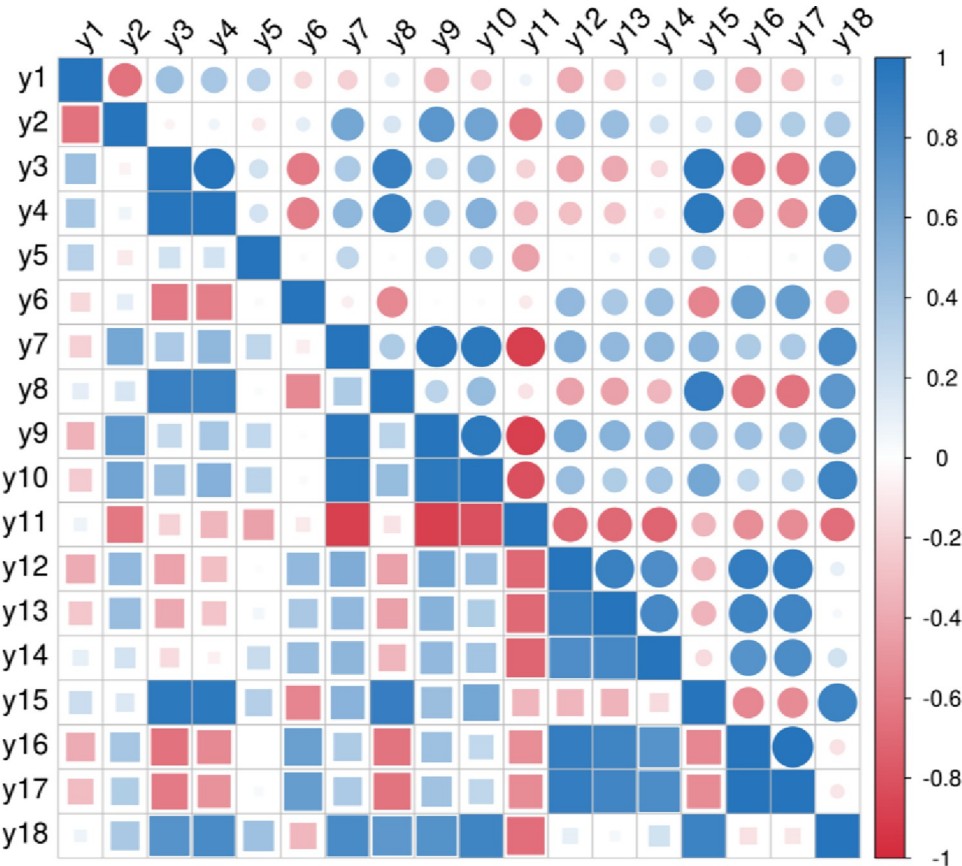

**Fig 2. Correlation analysis of Pearson coefficient between the characteristics of rural revitalization dimension.**

comprehensive measurement of the system by indicator [49]. A comprehensive evaluation model usually involves multiple indicators and factors; in this process, the selection of weights occupies an important position, and the entropy value method can objectively assign values to the indicators of each subsystem and reduce subjective errors [49]. The entropy value method can effectively measure the differences between the indicators and convert the differences and importance of the indicators into figures, providing a scientific and quantitative basis for comprehensive evaluation. First, the data are standardized. As can be seen from Table 1, the rural revitalization, new-type urbanization, and digital economy systems all contain multiple evaluation indicators. Given the variations in unit, magnitude, and direction of influence among the indicators, standardization is essential to mitigate the impact of these differences on the analysis [50]. The standardized formulas used are as follows:

For positive indicators:

$$R_{ij} = \frac{x_{ij} - \min(x_{ij})}{\max(x_{ij}) - \min(x_{ij})} \tag{1}$$

For negative indicators:

$$R_{ij} = \frac{\max(x_{ij}) - x_{ij}}{\max(x_{ij}) - \min(x_{ij})} \tag{2}$$

where $i = 1,2,\ldots n$ represents the year, $j = 1,2,\ldots,m$ denotes the indicator, $x_{ij}$ is the initial value

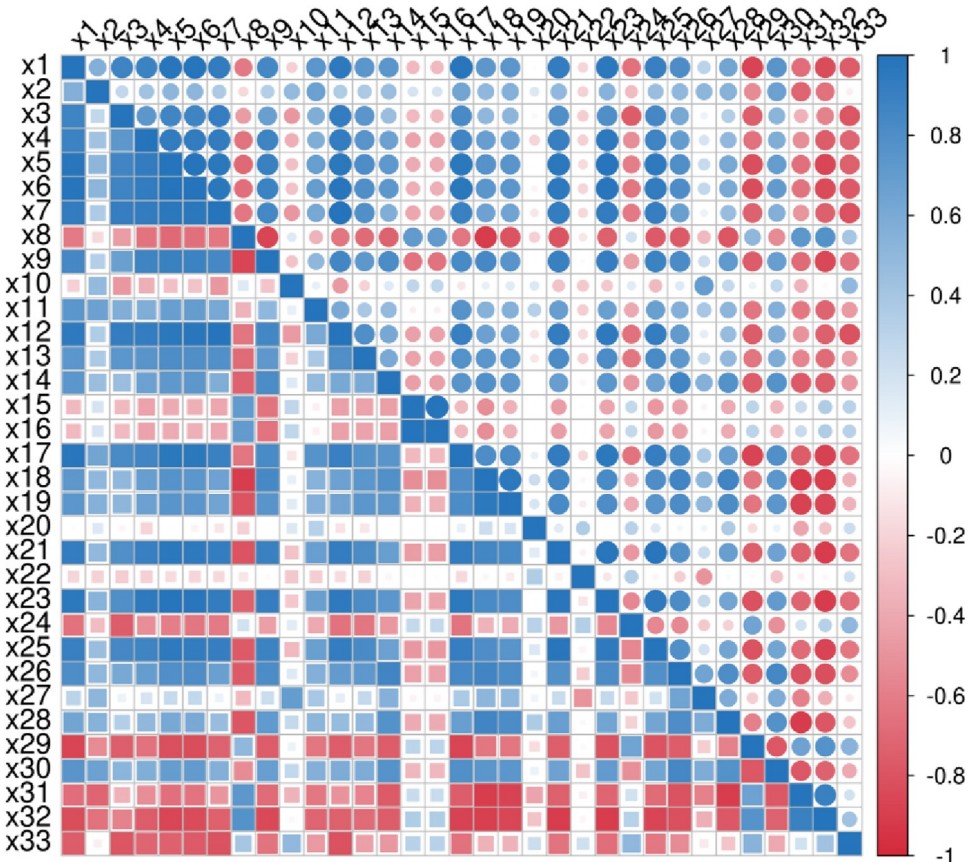

**Fig 3.** Pearson coefficient correlation analysis between the characteristics of the new urbanization dimension.

of the indicator $j$ in the year $i$, $\max(x_{ij})$ and $\min(x_{ij})$ denotes the maximum and minimum values of the $jth$ indicator, respectively. $R_{ij}$ represents the standard value after dimensionless treatment $x_{ij}$.

Regarding the principle of the entropy method, first, the weight $P_{ij}$ of the indicator $j$ in the year $i$ denotes calculated based on the standardized indicator value; second, the information entropy value $E_i$ of indicator $j$ is calculated; finally, the weight $W_j$ of indicator $j$ is calculated, which is shown in Eqs (3)–(5).

$$P_{ij} = \frac{R_{ij}}{\sum_{i=1}^{n} R_{ij}} \tag{3}$$

$$E_j = -(1/\ln n)\sum_{i=1}^{n} P_{ij}\ln P_{ij} \tag{4}$$

$$W_j = (1 - E_j)/\sum_{j=1}^{m}(1 - E_j) \tag{5}$$

Finally, the combined evaluation level of the three subsystems was calculated.

$$U_x = \sum_{i}^{m} W_{ij}R_{ij}, \quad x = 1, 2, 3 \tag{6}$$

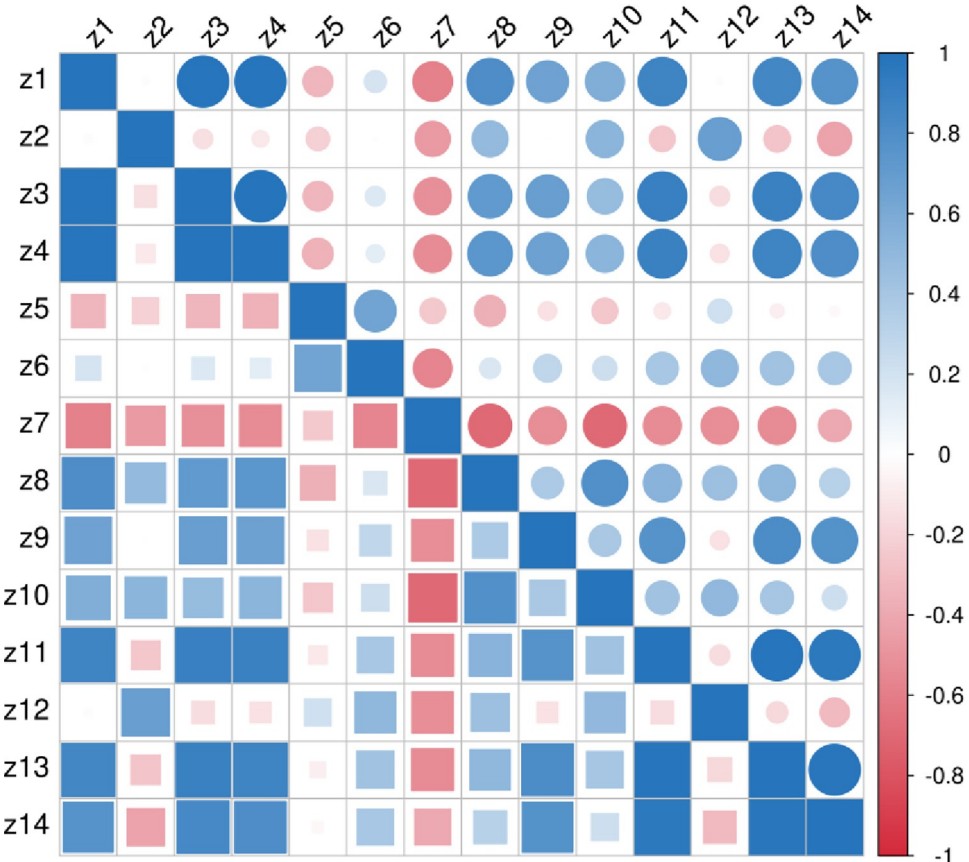

**Fig 4. Pearson coefficient correlation analysis of the digital economy characteristics.**

$$\sum\nolimits_{i=1}^{m} W_{ij} = 1 \tag{7}$$

where $U_1$, $U_2$, and $U_3$ are the combined evaluation values for the digital economy, rural revitalization, and new-type urbanization systems, respectively.

**2.2.2 Evaluation model construction of the coupling coordination degree.** The three systems of the digital economy, rural revitalization, and new-type urbanization are closely linked, so the coupling degree, a physics category, is used to explore the dynamic relationship between the three systems in terms of interaction and interdependence. The coupling degree reveals the degree of interaction among the three subsystems, and the formula is as follows:

$$C = \frac{3\sqrt[3]{U_1 U_2 U_3}}{U_1 + U_2 + U_3} \tag{8}$$

The coupling degree can reflect the strength of the interaction between the systems, but it cannot reflect the overall coordination between the systems; for example, when the integrated development level of the digital economy, rural revitalization, and new-type urbanization is low, its coupling degree is still high, at this time, the value of the coupling degree cannot respond to the overall development and synergy effect between the three systems [49]. Therefore, a coupled coordination degree model needs to be introduced to objectively reflect the

level of coordinated development of the three systems [51]. The formula is as follows:

$$T = \alpha U_1 + \beta U_2 + \delta U_3 \tag{9}$$

$$D = \sqrt{C \times T} \tag{10}$$

where $C$ denotes the coupling degree of the three systems of rural revitalization, new-type urbanization, and digital economy, whose value ranges from [0, 1], and the value of the coupling degree is positively proportional to the degree of association between the systems. $T$ is the comprehensive evaluation level of the three systems of rural revitalization, new-type urbanization, and digital economy. In the process of rural and urban development, rural revitalization, new-type urbanization, and the digital economy are in an equally important position. Therefore, the three values of α, β, and δ are set to be the same, and the sum of the three numbers is 1. $D$ is the coupling coordination degree of the three systems of rural revitalization, new-type urbanization, and digital economy. Referring to the classification method of scholars [52], the coupling coordination degree and coupling degree type are divided (Table 2).

**2.2.3 Obstacle degree model.** In order to effectively improve and enhance the coupling coordination of the RR-NU-DE composite model system, it is necessary to explore the main obstacle factors affecting the model, analyses the influencing factors, and propose corresponding adjustment measures, which can help to achieve digitalization and sustainable development of urban and rural areas. Based on the barrier degree model, the index deviation is introduced to diagnose the obstacle factors of the RR-NU-DE conformity model [53]. The equation is as follows.

Calculation of the indicator deviation:

$$D_{ij} = 1 - R_{ij} \tag{11}$$

(2) Calculation of the obstacle degree indicators:

$$O_i = (D_{ij} w_i / \sum_{j=1}^{n} D_{ij} w_i) \times 100\% \tag{12}$$

where $O_i$ represents the degree obstacle of the indicators; $w_i$ is the indicator weight; $D_{ij}$ is the indicator deviation; and $R_{ij}$ is the normalized value.

**Table 2. Coupling coordination degree level and coupling degree type classification are established.**

| CCD | CCD level | CCD state | CCD type |
|---|---|---|---|
| 0.0–0.09 | Extremely unbalanced development | Unbalanced development | Lower level |
| 0.1–0.19 | Seriously unbalanced development | | |
| 0.2–0.29 | Moderately unbalanced development | | |
| 0.3–0.39 | Slightly unbalanced development | | Low level |
| 0.4–0.49 | Barely unbalanced development | Semi-balanced development | |
| 0.5–0.59 | Barely balanced development | | High level |
| 0.6–0.69 | Slightly balanced development | Balanced development | |
| 0.7–0.79 | Moderately balanced development | | |
| 0.8–0.89 | Favorably balanced development | | Higher level |
| 0.9–1.0 | Superiorly balanced development | | |

## 2.3 Data sources and preprocessing

This research is based on 31 provinces across the country; Hong Kong, Macau, and Taiwan are excluded due to the lack of data on some indicators. In order to ensure the consistency of the data, the time of the study is unified as 2011–2021. The original data for the study were obtained from the China Statistical Yearbook (2011–2021), the China Urban and Rural Construction Statistical Yearbook (2011–2021), provincial city yearbooks, and the EPS data platform. Due to some missing data, the interpolation method was used to fill in and make up for the shortcomings of the simple average method. Due to historical and locational factors, relevant data are missing in some regions, such as Xinjiang and Tibet, and this study uses the interpolation method to fill in and make up for the shortcomings of the simple average method.

# 3. Result analysis

## 3.1 Analysis of the comprehensive development level of the RR-NU-DE system

As can be seen from Fig 5, the comprehensive development level of rural revitalization, new-type urbanization, and digital economy in 2011–2021 shows an overall upward trend, with the comprehensive evaluation index increasing from 0.2022 in 2011 to 0.2978 in 2021, with the value increasing year by year. In sum, the comprehensive development level can be divided into two stages: in the first stage (2011–2015), the comprehensive development level of rural revitalization, new-type urbanization, and the digital economy are all on an upward trend, so the comprehensive development level is developing rapidly; in the second stage (2016–2019),

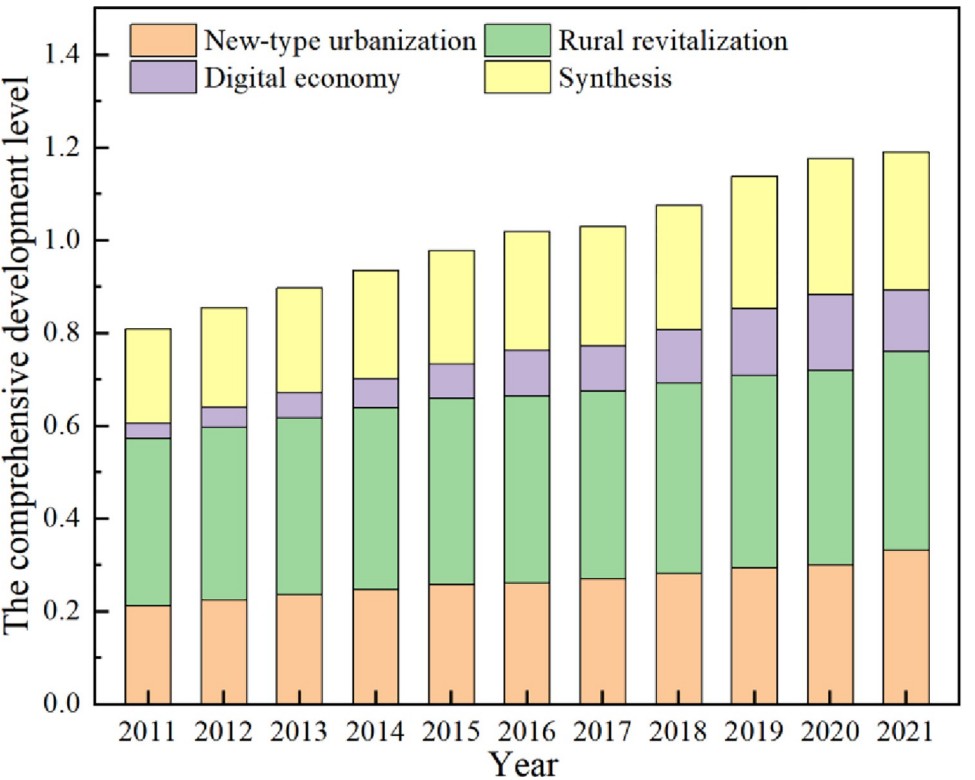

**Fig 5. The comprehensive development level of the RR-NU-DE composite system.**

the comprehensive development level of new-type urbanization and rural revitalization is slowly rising, and the digital economy is growing rapidly, so the level of the comprehensive development is stable; in the third stage (2020–2021), then the level of the digital economy development is declining, and rural revitalization and new-type urbanization are rising in general, which ultimately leads to a smaller increase in the comprehensive development level.

In terms of the digital economy, the comprehensive evaluation index grew from 0.0332 in 2011 to 0.1626 in 2020. The national level of digital economy development in 2012 shows a growing trend inseparable from the digital environment's expansion and the scale of digital applications. The rapid development of the digital economy provides sufficient information support for urban construction and provides data coordination between the development of urban systems [54]. There is an upward trend from 2011 to 2021 in terms of new-type urbanization. The index of new-type urbanization was the smallest in 2011, only 0.2126, and has already reached 0.3325 in 2021. As seen in Fig 5, the most rapid development of new-type urbanization across the country will be in 2012–2013 and 2020, during which the country actively develops new urbanization and promotes urban and rural construction.

Regarding rural revitalization, the composite evaluation index of rural revitalization is the smallest in 2011 and the largest in 2021. Compared with the digital economy and new-type urbanization, the development level of rural revitalization fluctuates the least and has been higher than both, showing a slow upward trend. As shown in Fig 5, the growth rate decreased in 2015 but increased in 2018. The three subsystems of rural revitalization, new-type urbanization, and the digital economy have more obvious characteristics. From 2011 to 2020, the comprehensive evaluation index of rural revitalization has always been greater than that of new-type urbanization and digital economy, and the comprehensive evaluation index of digital economy has always been at the lowest level. This indicates that China's new-type urbanization and rural revitalization have made great progress, but the digital economy has developed but lagged behind the development level of new-type urbanization and rural revitalization.

The 31 provinces continue to be subdivided into six regions based on the division of China's geographic zones into East, Central, and West [55], Northeast: Liaoning, Heilongjiang, Jilin; Southeast: Jiangsu, Zhejiang, Shanghai, Fujian, Shandong, Guangdong, Hainan; North China: Beijing, Tianjin, Inner Mongolia, Hebei, Shanxi; Central China: Anhui, Henan, Hunan, Hubei, Jiangxi; Northwest: Shaanxi, Gansu, Qinghai, Ningxia, Xinjiang; Southwest China: Yunnan, Guizhou, Sichuan, Chongqing, Tibet, Guangxi. Changes in each region's integrated development level are presented in Fig 6.

The comprehensive development level of the three provinces in the Northeast region has continued to rise, and the three provinces have shown parallel development (Fig 6(A)). From 2011 to 2021, the gap between Jilin Province and the other two provinces gradually narrowed, surpassing the other two in the comprehensive development level in 2016. Jilin's border with Russia, its distance from the economic center, the backward development of the digital economy, and the slow development of rural revitalization have led to the relative backwardness of the comprehensive development level.

As can be seen from the Fig 6(B), In 2011, the highest comprehensive development level index of Jiangsu Province in the southeast region was Shandong Province, followed by Jiangsu Province. In 2021, the comprehensive development level changed dramatically, and the highest development index was Guangdong Province, followed by Jiangsu Province. Guangdong Province ranked first in the country in terms of the number of 5G digital base stations, and the scale of digital economic growth ranked first for five consecutive years. The rapid development of the digital economy has led to the revitalization of the countryside at the same time, and first-class scientific and technological research and development technology and personnel

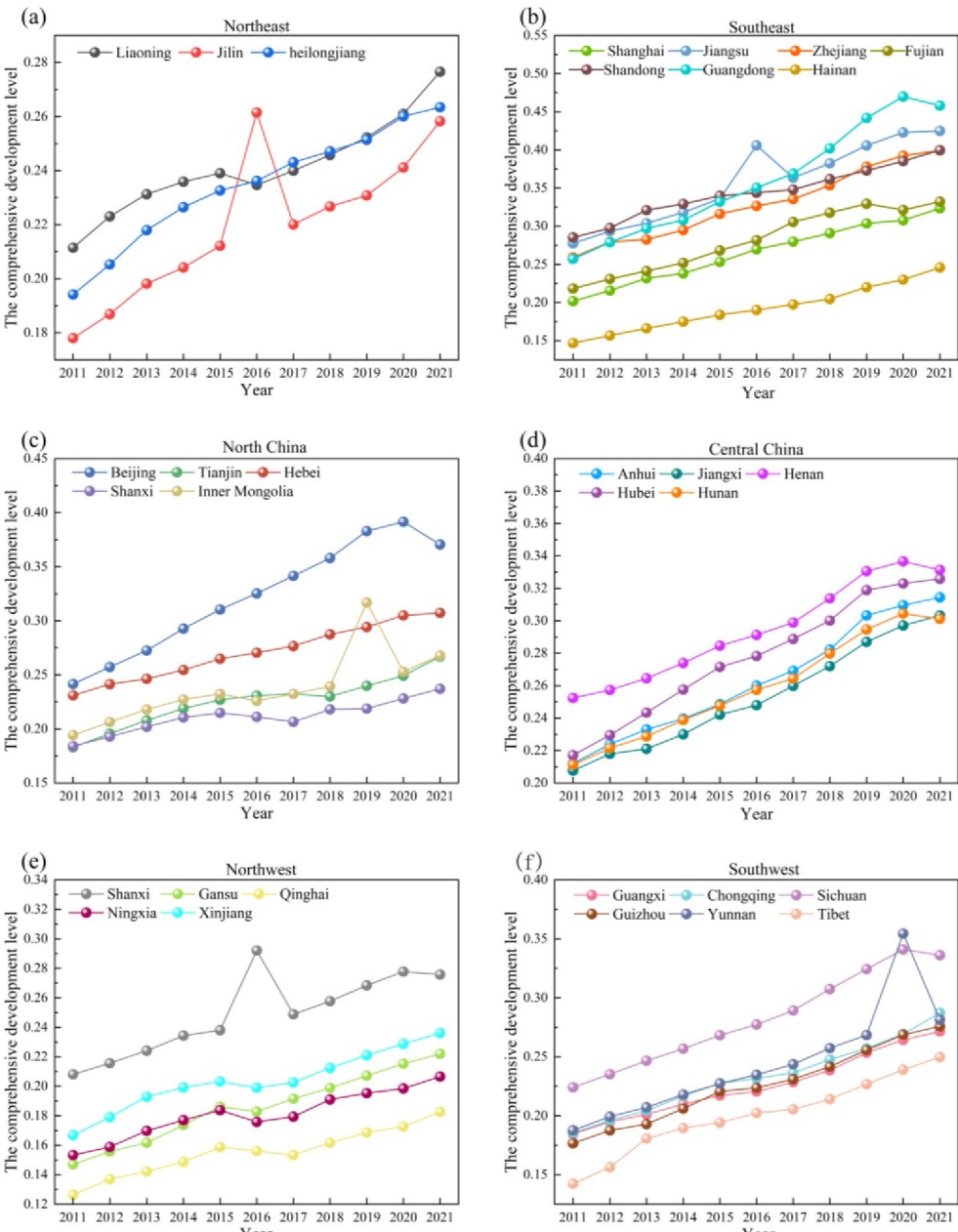

**Fig 6. The comprehensive development level of the RR-NU-DE composite system in all regions.**

have promoted the transformation of rural agriculture and the realization of the revitalization of the countryside as well as the promotion of new-type urbanization.

As can be seen from Fig 6(C), the comprehensive development level of Beijing has been in the first place, and the Inner Mongolia Autonomous Region has been in the last place. Inner Mongolia Autonomous Region has a vast territory, is a significant energy export province, a vital livestock base, rapid development of rural revitalization, and relatively backward development of the digital economy, mainly due to the gradual increase of the population outflow area in Inner Mongolia, which has hindered the expansion of its digital economic development. As seen from Fig 6(D), Central China has a relatively balanced level of comprehensive

development, of which Henan Province has always been at the forefront. As a major agricultural province, Henan Province has been actively promoting the return of forests to the countryside in recent years. It has made revitalizing the countryside one of its "ten major development strategies." Henan Province actively promotes rural revitalization in terms of industry, talent, ecology, culture, and organizational development.

As seen from Fig 6(E), Shaanxi Province has always been at the forefront of the comprehensive development level index in the Northwest region, and Qinghai Province has always been in the last position. Shaanxi Province is closer to the economic center than the other provinces. Qinghai Province is located in the northwest of China, on a plateau, and the geographical environment has hindered the development of the digital economy to a relative degree; in addition, the product structure of Qinghai Province is single, with traditional industries as the mainstay, and there is less demand for and application of digital technology, which limits the development of the digital economy to a certain extent. As can be seen from Fig 6(F), the gap between the comprehensive development level of Chongqing Municipality and Yunnan Province in the southwest region is relatively small, presenting a situation of neck-and-neck progress, while the comprehensive development level of the Tibet Autonomous Region has been at the end. Chongqing Municipality and Yunnan Province vigorously develop tourism to drive economic development. Among them, Yunnan Province vigorously promotes rural tourism development and actively encourages the integration of culture and tourism, as well as rural tourism development, to drive rural revitalization.

## 3.2 Analysis of the coupled coordination degree of the RR-NU-DE system

**3.2.1 Analysis of the temporal evolution of the coupled coordination degree of the RR-NU-DE system.** As shown in Fig 7, the coupling degree of rural revitalization, new-type urbanization, and digital economy in the country is maintained above 0.6 from 2011 to 2021, and the coupling degree is between 0.7 and 0.9 in most years, showing an increasing trend year by year. It can be seen that the degree of coupling among rural revitalization, new-type urbanization, and the digital economy is high level. This phenomenon is because the three major subsystems of rural revitalization, new-type urbanization, and digital economy have all been improved from 2011 to 2021, pushing the coupling degree to increase rapidly, and the degree of coupling coordination is on an upward trend. The coupling coordination level transitions from slightly unbalanced to barely balanced development, growing from 0.3696 in 2011 to 0.5232 in 2020, with a growth rate of 41.57%. During the study period, the 19th National Congress formally put forward the rural revitalization strategy, which addresses the "three rural issues" and promotes rural revitalization by advancing agricultural supply-side reform, deepening rural reform, and developing rural characteristics. In terms of new-type urbanization, the government has launched a series of relevant policies, such as promoting the transfer of the agricultural population to citizenship, strengthening employment services for farmers, improving supporting facilities, and developing small towns with special characteristics to promote the process of new-type urbanization. Concerning the digital economy, over the years, various provinces have made great efforts to develop the digital economy, such as accelerating the construction of network infrastructures, establishing a system of laws and regulations on the digital economy, and strengthening international cooperation to promote the globalization of the digital economy, thus laying a solid foundation for the development of the digital economy.

From the results of the evolution of individual provinces across the country from 2011 to 2021, the coupled coordination level of the rural revitalization, new-type urbanization, and digital economy system in most provinces is in the transition stage of slightly unbalanced and

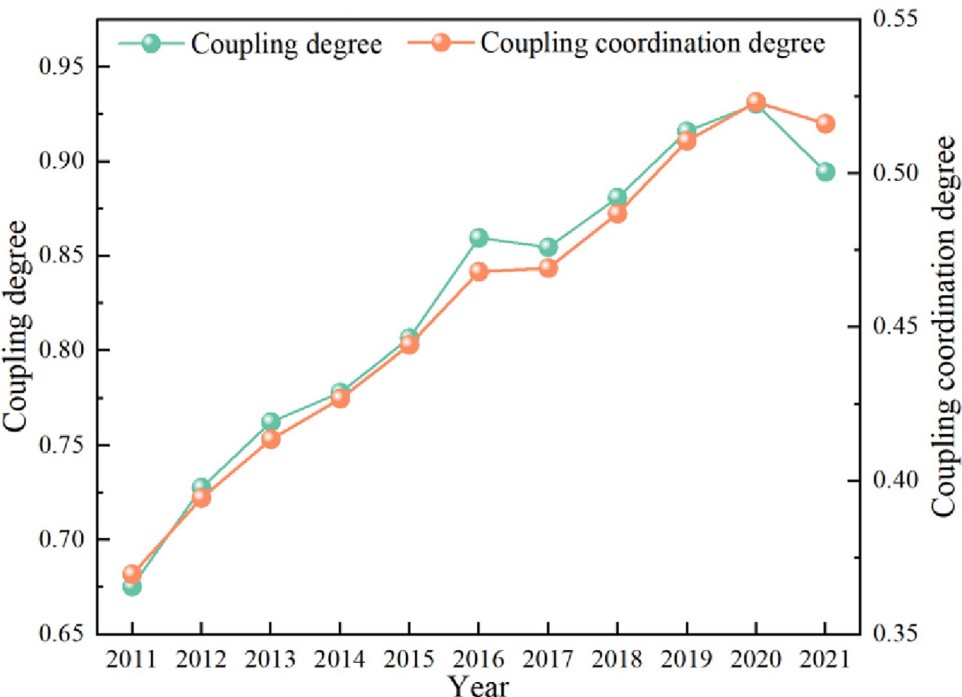

**Fig 7. The coupling coordination degree of the RR-NU-DE composite system.**

barely balanced development, with only a few provinces in the slightly balanced development stage. Five provinces have reached a slightly balanced development: Beijing, Jiangsu, Shandong, Guangdong, and Zhejiang. Beijing grows from 0.4560 in 2011 to 0.6141 in 2020, Jiangsu Province from 0.4907 in 2011 to 0.6508 in 2021, Zhejiang Province from 0.4566 in 2011 to 0.6271 in 2021, Shandong Province from 0.4569 in 2011 to 0.6192 in 2021, Guangdong Province grows from 0.4766 in 2011 to 0.6791 in 2020, and the type of coordination improves from slightly unbalanced to slightly unbalanced development. In addition, the provinces in the unbalanced development stage from 2011 to 2021 include Ningxia, which grew from 0.2565 to 0.3724, and Qinghai Province, which grew from 0.2363 in 2011 to 0.3499 in 2021. All of these provinces are cities in western China, facing multiple development pressures such as rural upgrading and transformation, lagging in the development of the digital economy, and lagging in the ecological environment, which leads to these provinces with low coupling and coordination of digital economy, new-type urbanization, and rural revitalization systems, and slow growth.

**3.2.2 Analysis of the spatial evolution of the coupled coordination degree of the RR-NU-DE system.** To see more clearly and intuitively the spatial evolution characteristics of the national rural revitalization, new-type urbanization, and digital economy system, this research selected 2011, 2016, and 2021 as the representative years and presented the spatial visualization of the coupled coordination degree of rural revitalization, new-type urbanization, and digital economy system of the 31 provinces in China using ArcGIS software.

As shown in Fig 8. From the perspective of China's 31 provinces, the coupling coordination degree of the RR-NU-DE system from 2011 to 2021 grows over time, developing from a moderately, slightly unbalanced development situation in 2011 to a barely balanced, slightly balanced development situation in 2021, which shows that the coupling coordination of each province's RR-NU-DE system towards benign development.

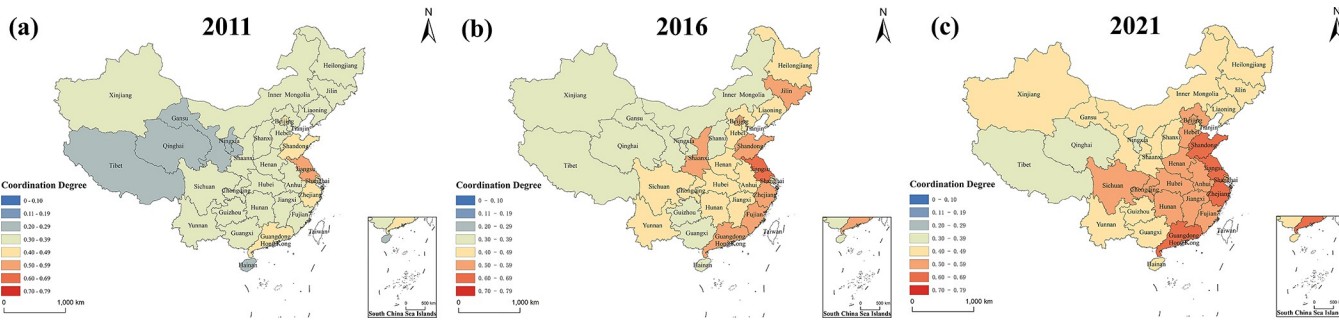

**Fig 8. The coupling coordination degree of China's RR-NU- DE composite system.**

As shown in Fig 9. From the perspective of the provinces of China, the coupled coordination of the RR-NU-DE system shows significant spatial variability across provinces. Southeast areas such as Shandong Province, Jiangsu Province, Shanghai, Zhejiang Province, and Guangdong Province are at the absolute leading level, of which Jiangsu Province is the first to enter the slightly balanced development in 2016, followed by other provinces in 2021 also entered this stage. In 2011, the spatial regions showed three kinds of coupling coordination situations: Xinjiang, Qinghai, and Tibet in the western region were moderately unbalanced development, Shanxi in the northern region, Sichuan and Yunnan in the southwestern region were slightly unbalanced development, and the southeast areas such as Shandong Province, Jiangsu Province, and Shanghai City were on the barely unbalanced development. Compared to the coupling coordination level in 2011, in 2016, the coupling coordination level in western regions evolved from moderately unbalanced development to slightly unbalanced development, the coupling coordination level in Central and North China regions evolved from slightly unbalanced development to barely unbalanced development, and the coupling coordination level in Southeast area evolved from barely unbalanced development to barely balanced development, with Jiangsu Province directly growing from barely unbalanced development to slightly balanced development. In 2021, the coupling coordination level evolved from slightly unbalanced development to barely unbalanced development in all provinces in the western region except Qinghai Province, Tibet, and Ningxia, showing no change. The central region grows from barely unbalanced development to barely balanced development. The Southeast region grows from barely balanced development to moderately balanced development. In all three time cross-sections, provinces with areas of moderately unbalanced development in 2011 reached more than slightly unbalanced development in 2021.

In order to explore more intuitively the changes in the spatial pattern of the coupling coordination degree of each region, the coupling coordination degree of three provinces in each region in 2011, 2016, and 2021 were selected, and the spatial changes in the coupling coordination degree of each region were plotted as follows.

As shown in Fig 9, the coupling coordination degree has changed the most. The province with the most changes in the region is the Southeast region, covering four coupling coordination degree levels, which has developed from the initial moderate dysfunctionality to primary coordination. The degree of coupling coordination of each province in the region has changed and gradually developed in the direction of primary coordination. Especially in Jiangsu Province, it developed directly from near disorder in 2011 to primary coordination in 2016. Jiangsu Province implements digital farms led by digital innovations to cover the province's agricultural and rural areas with digital-featured industrial chains, promotes rural revitalization with the development of the digital economy; and promotes tourism development through the

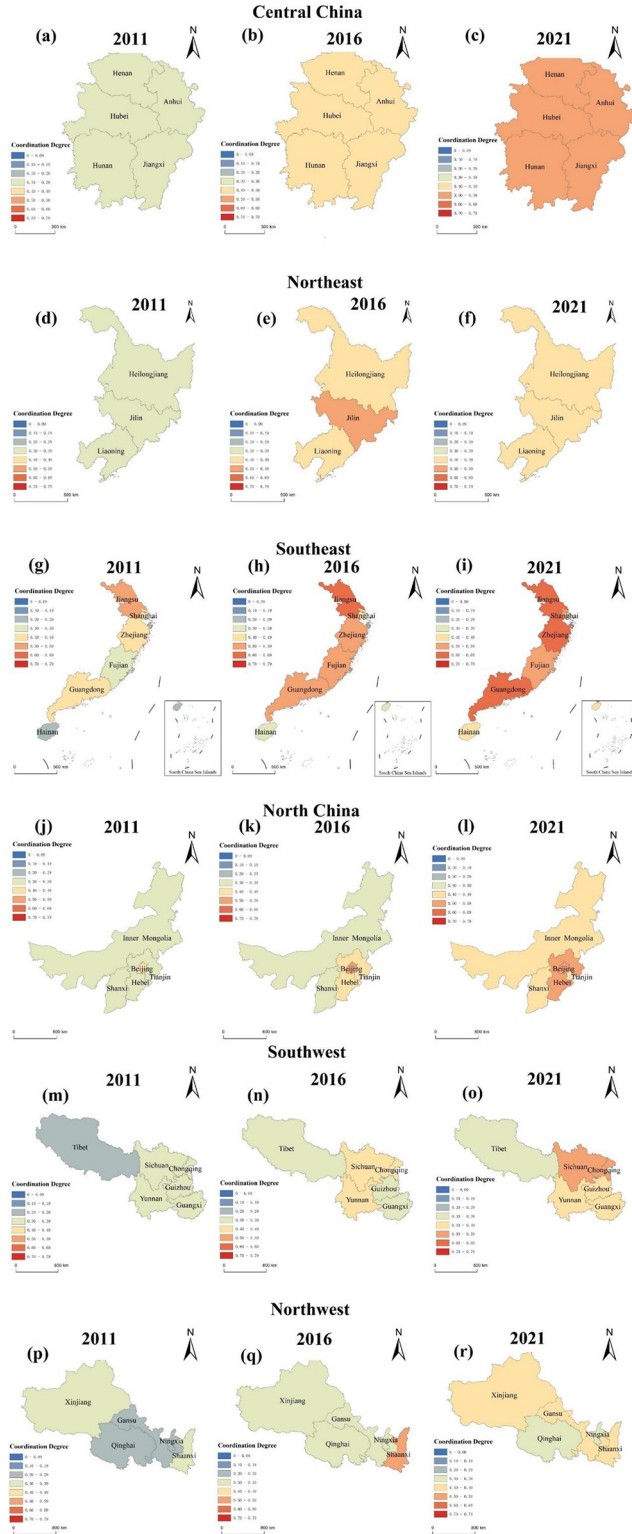

**Fig 9. The coupling coordination degree of the RR-NU- DE systems in China's region (Note: Based on the standard map with review number GS (2016)1595 on the standard map service website of the Ministry of Natural Resources, with no modifications to the base map).**

preservation of historical and cultural villages, etc., which injects vitality into the countryside and promotes the growth of new-type urbanization. The changing trend is not apparent in the Northeast, where the coupling coordination rank is on the verge of dislocation in 2016 and 2021. The three northeastern provinces have a high proportion of traditional industries and the relatively slow development of new industries and modern technology, which limits the development of the digital economy, and the lack of backbone in technological innovation capacity, and to a certain extent, hinders the promotion of rural revitalization and urban modernization.

## 3.3 Analysis of the obstacle factors of the RR-NU-DE system

In order to further explore which factors hinder the coupling coordination level of rural revitalization, new-type urbanization, and digital economy system, Through Eq (11) and Eq (12), the indicator deviation and barrier are calculated, respectively. The obstacle degree model is used to explore the obstacles to the differences in the coupling coordinated development of the 31 provinces across the country in 2011–2021. The final obstacle degree of each province is calculated by calculating the mean value of the obstacle degree of each index, and the top three obstacle factors are defined as the main obstacle factors. In order to have a clearer understanding of whether the degree of obstacles in each province will change over time, the top 5 obstacle factors were calculated for the three years 2011, 2016, and 2021 (see Table 3 and Fig 10).

From the perspective of the new-type urbanization system, the main obstacle factors for all provinces except Beijing, Tianjin, Shanghai, Jiangsu, Tibet, and Qinghai include per capita local financial income (x7), per capita mileage of road lines owned (X8), and per capita local financial expenditure on science and technology (X12), which indicates that the economy, infrastructure, and science and technology are the obstacles to the new-type urbanization of all provinces. Per capita local fiscal revenue is an essential indicator of economic development and one of the measures of market economic vitality. Lagging economic development hinders the urbanization process to a certain extent; the construction of infrastructure, such as road construction, is one of the most critical indicators for promoting rural and urban development, and poor infrastructure hinders the link between the countryside and the city, thus restricting the urbanization process. Per capita local financial expenditure on science and technology represents the importance attached to science and technology, and scientific and technological innovation is the driving force behind new-type urbanization, which profoundly impacts how new-type urbanization develops and evolves.

In terms of the rural revitalization system, the main obstacle factors for each province, the number of rural doctors and health workers (y8) and the number of illiterate people/population aged 15 and above (y11), indicate that the medical situation in the countryside and the literacy level of the rural population are high obstacles to rural revitalization. The number of rural doctors and sanitarians directly reflects the level of medical care in villages. An increase in the number of rural doctors and sanitarians can provide more medical services in the countryside. Good health is the basis for wealth creation and economic development for farmers, and an increase in the number of rural doctors and sanitarians can help to raise the level of medical services in rural areas, improve the situation of healthcare resources, and promote the process of rural revitalization. This is one of the reasons why the State attaches importance to developing rural health in its rural revitalization strategy. There is a close relationship between the number of illiterates and rural revitalization. Rural revitalization aims to achieve prosperity in the rural economy and comprehensive development of farmers, and the problem of illiteracy poses a major obstacle to achieving this goal. The illiterate population's lack of basic literacy and knowledge limits their ability to participate in modern agricultural production and

**Table 3. Ranking of main obstacle factors in the index layer of selected provinces(cities).**

| Regions | Provinces (cities) | Item | NE subsystem | | | RR subsystem | | | DE subsystem | | |
|---|---|---|---|---|---|---|---|---|---|---|---|
| | | | Ranking of main obstacle factors (%) | | | | | | | | |
| | | | 1st | 2rd | 3th | 1st | 2rd | 3th | 1st | 2rd | 3th |
| North China | Beijing | Obstacle factor | x8 | x11 | x14 | y11 | y8 | y15 | z7 | z3 | z1 |
| | | Obstacle degree | 16.64 | 12.26 | 7.98 | 26.85 | 22.84 | 9.92 | 28.71 | 10.03 | 9.54 |
| | Tianjin | Obstacle factor | x12 | x8 | x3 | y11 | y8 | y5 | z7 | z2 | z8 |
| | | Obstacle degree | 12.26 | 12.16 | 8.39 | 25.59 | 21.98 | 11.91 | 23.98 | 12.13 | 10.79 |
| | Hebei | Obstacle factor | x12 | x8 | x7 | y11 | y8 | y5 | z7 | z2 | z8 |
| | | Obstacle degree | 15.11 | 9.57 | 9.15 | 37.55 | 14.94 | 13.25 | 24.51 | 13.12 | 11.68 |
| | Shanxi | Obstacle factor | x12 | x8 | x7 | y11 | y8 | y5 | z7 | z2 | z8 |
| | | Obstacle degree | 14.70 | 9.10 | 8.67 | 29.09 | 18.66 | 11.45 | 23.59 | 12.81 | 11.34 |
| | Inner Mongolia | Obstacle factor | x12 | x7 | x8 | y11 | y8 | y5 | z7 | z2 | z8 |
| | | Obstacle degree | 15.67 | 8.37 | 8.35 | 27.77 | 22.71 | 11.13 | 21.32 | 13.11 | 11.55 |
| Northeast | Liaoning | Obstacle factor | x12 | x8 | x7 | y11 | y8 | y5 | z7 | z2 | z8 |
| | | Obstacle degree | 14.95 | 9.87 | 8.75 | 31.08 | 22.46 | 7.13 | 24.46 | 12.74 | 10.55 |
| | Jilin | Obstacle factor | x12 | x8 | x7 | y11 | y8 | y1 | z7 | z2 | z8 |
| | | Obstacle degree | 14.66 | 9.09 | 8.98 | 30.84 | 23.98 | 6.84 | 22.22 | 12.70 | 11.21 |
| | Heilongjiang | Obstacle factor | x12 | x7 | x8 | y11 | y8 | y5 | z7 | z2 | z8 |
| | | Obstacle degree | 14.85 | 9.30 | 8.83 | 33.11 | 24.90 | 6.51 | 23.55 | 12.63 | 11.28 |
| Southeast | Shanghai | Obstacle factor | x8 | x11 | x3 | y11 | y8 | y5 | z7 | z2 | z8 |
| | | Obstacle degree | 14.84 | 10.54 | 7.77 | 24.76 | 21.95 | 11.21 | 25.49 | 12.25 | 9.73 |
| | Jiangsu | Obstacle factor | x12 | x8 | x11 | y11 | y8 | y5 | z7 | z2 | z10 |
| | | Obstacle degree | 14.23 | 11.73 | 8.86 | 32.87 | 23.58 | 15.30 | 29.07 | 15.22 | 9.50 |
| | Zhejiang | Obstacle factor | x12 | x8 | x7 | y11 | y8 | y1 | z7 | z2 | z8 |
| | | Obstacle degree | 13.48 | 11.18 | 8.58 | 33.41 | 30.16 | 10.14 | 28.76 | 14.84 | 11.02 |
| | Fujian | Obstacle factor | x12 | x8 | x7 | y11 | y8 | y1 | z7 | z2 | z8 |
| | | Obstacle degree | 14.87 | 10.13 | 8.86 | 32.21 | 25.84 | 7.52 | 25.58 | 13.89 | 11.12 |
| | Shandong | Obstacle factor | x12 | x8 | x7 | y11 | y5 | y1 | z7 | z2 | z8 |
| | | Obstacle degree | 15.55 | 10.43 | 9.43 | 39.44 | 18.35 | 10.01 | 27.65 | 13.83 | 10.57 |
| | Guangdong | Obstacle factor | x12 | x8 | x7 | y11 | y8 | y1 | z7 | z2 | z8 |
| | | Obstacle degree | 13.21 | 11.34 | 8.86 | 34.89 | 25.26 | 10.00 | 34.69 | 15.59 | 8.91 |
| | Hainan | Obstacle factor | x12 | x8 | x7 | y11 | y8 | y15 | z7 | z2 | z8 |
| | | Obstacle degree | 14.50 | 9.50 | 8.40 | 27.24 | 24.82 | 9.38 | 23.04 | 12.62 | 11.05 |
| Central China | Anhui | Obstacle factor | x12 | x8 | x7 | y11 | y8 | y5 | z7 | z2 | z8 |
| | | Obstacle degree | 13.24 | 9.67 | 9.38 | 31.41 | 21.58 | 11.47 | 24.66 | 13.00 | 11.66 |
| | Jiangxi | Obstacle factor | x12 | x8 | x7 | y11 | y8 | y1 | z7 | z2 | z8 |
| | | Obstacle degree | 14.15 | 9.30 | 9.09 | 34.55 | 21.83 | 9.06 | 23.93 | 12.94 | 11.46 |
| | Henan | Obstacle factor | x12 | x8 | x7 | y11 | y5 | y1 | z7 | z2 | z8 |
| | | Obstacle degree | 14.93 | 9.68 | 9.57 | 40.25 | 15.66 | 10.36 | 24.99 | 13.47 | 11.87 |
| | Hubei | Obstacle factor | x12 | x8 | x7 | y11 | y8 | y5 | z7 | z2 | z8 |
| | | Obstacle degree | 14.00 | 9.48 | 9.44 | 33.92 | 23.33 | 8.52 | 24.95 | 12.00 | 11.26 |
| | Hunan | Obstacle factor | x12 | x8 | x7 | y11 | y8 | y1 | z7 | z2 | z8 |
| | | Obstacle degree | 14.64 | 9.29 | 9.28 | 35.15 | 21.69 | 8.27 | 24.61 | 13.04 | 11.56 |

(*Continued*)

**Table 3.** (Continued)

| Regions | Provinces (cities) | Item | NE subsystem | | | RR subsystem | | | DE subsystem | | |
|---|---|---|---|---|---|---|---|---|---|---|---|
| | | | Ranking of main obstacle factors (%) | | | | | | | | |
| | | | 1st | 2rd | 3th | 1st | 2rd | 3th | 1st | 2rd | 3th |
| Northwest | Shanxi | Obstacle factor | x12 | x8 | x7 | y11 | y8 | y1 | z7 | z2 | z8 |
| | | Obstacle degree | 15.59 | 9.55 | 9.32 | 29.22 | 21.15 | 7.37 | 22.88 | 11.85 | 11.71 |
| | Gansu | Obstacle factor | x12 | x7 | x8 | y11 | y8 | y5 | z7 | z2 | z8 |
| | | Obstacle degree | 14.75 | 9.32 | 8.41 | 22.48 | 20.75 | 12.82 | 23.29 | 12.50 | 11.21 |
| | Qinghai | Obstacle factor | x12 | x7 | x11 | y8 | y11 | y5 | z7 | z2 | z8 |
| | | Obstacle degree | 14.67 | 9.17 | 6.25 | 19.98 | 17.77 | 12.15 | 22.92 | 12.48 | 11.02 |
| | Ningxia | Obstacle factor | x12 | x8 | x7 | y8 | y11 | y5 | z7 | z2 | z8 |
| | | Obstacle degree | 13.66 | 8.94 | 8.79 | 21.61 | 21.44 | 11.41 | 22.95 | 12.56 | 11.02 |
| | Xinjiang | Obstacle factor | x12 | x7 | x8 | y11 | y8 | y5 | z7 | z2 | z8 |
| | | Obstacle degree | 14.68 | 8.77 | 8.03 | 27.52 | 22.58 | 14.85 | 23.32 | 12.76 | 11.22 |
| Southwest | Guangxi | Obstacle factor | x12 | x8 | x7 | y11 | y8 | y1 | z7 | z2 | z8 |
| | | Obstacle degree | 14.25 | 9.12 | 9.03 | 33.47 | 22.86 | 6.74 | 23.74 | 12.94 | 11.31 |
| | Chongqing | Obstacle factor | x12 | x8 | x7 | y11 | y8 | y1 | z7 | z2 | z8 |
| | | Obstacle degree | 15.43 | 9.48 | 8.90 | 28.59 | 22.64 | 7.47 | 23.97 | 12.95 | 10.89 |
| | Sichuan | Obstacle factor | x12 | x7 | x8 | y11 | y8 | y5 | z7 | z2 | z8 |
| | | Obstacle degree | 14.50 | 9.01 | 9.00 | 34.41 | 17.77 | 9.72 | 25.60 | 12.97 | 10.71 |
| | Guizhou | Obstacle factor | x12 | x7 | x8 | y11 | y8 | y13 | z7 | z2 | z8 |
| | | Obstacle degree | 14.18 | 9.05 | 8.66 | 24.90 | 21.15 | 7.51 | 23.59 | 12.76 | 11.27 |
| | Yunnan | Obstacle factor | x12 | x7 | x8 | y11 | y8 | y13 | z7 | z2 | z8 |
| | | Obstacle degree | 14.80 | 9.08 | 8.57 | 29.10 | 22.12 | 7.50 | 21.46 | 13.30 | 11.75 |
| | Tibet | Obstacle factor | x12 | x7 | x11 | y8 | y5 | y15 | z7 | z2 | z8 |
| | | Obstacle degree | 15.35 | 9.53 | 8.21 | 22.53 | 12.75 | 11.07 | 23.00 | 12.59 | 10.36 |

rural economic development, and it also affects educational development and brain drain in rural areas.

In terms of the digital economic system, the digital financial coverage breadth index ($z7$), the total amount of technology contract turnover ($z2$), and software revenue ($z8$) are the main obstacle factors for all provinces except Beijing and Jiangsu. The digital financial coverage breadth index serves as a proxy for the accessibility of digital financial services within the digital economy era, thereby contributing to the resilience of the digital economy. The development of the digital economy provides space and opportunities for the breadth of digital financial coverage, improves the convenience and efficiency of financial services, and promotes the flow of capital and economic development. Technology contract transactions characterize the scale of the digital economy and are an important indicator of its vitality and innovation capacity. Software is an essential part of information technology, and the digital economy is formed based on information technology, so software revenue is an important indicator to measure the development of the digital economy. The software industry provides technical support for the digital economy.

In order to show the change in obstacle degree of each province from 2011 to 2021 more clearly, the barrier degree of 31 provinces is calculated for the times of 2011, 2016, and 2021, and the top five rankings of the barrier degree are retained as the main barrier factors. From the table, it can be found that the top five obstacle factors in each province have similarities. Three indicators, illiterate population/population aged 15 and above ($y11$), digital financial coverage breadth index ($z7$), and the number of rural doctors and health workers ($y8$), are

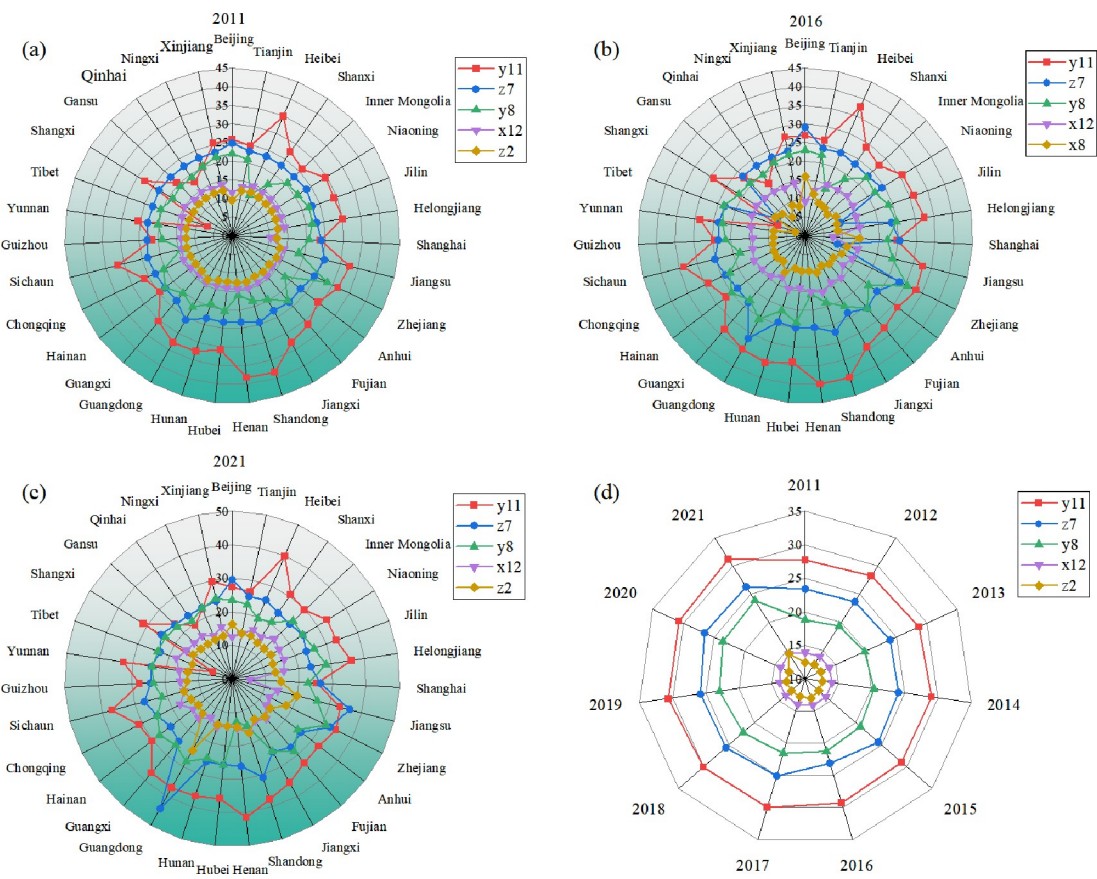

**Fig 10. The obstacle analysis of China's RR-NU-DE complex system.**

ranked in the top three of the main obstacle degree factors most of the time. Furthermore, the analysis identified per capita road mileage (x8) and the total value of technology contract transactions (z2) as significant impediments. There are 17 provinces whose main obstacle factors remain unchanged at all three-time points, which shows that the obstacles affecting rural revitalization-new-type urbanization, and digital economy have not changed and have been hindering the development of the provinces.

The top five major obstacle factors are screened out by measuring the obstacle degree of the three systems from 2011 to 2021. As shown in Fig 10, there is no apparent change in the major obstacle factors, but the total amount of technology contract turnover (z2) in 2021 rises from fifth place in 2020 to fourth place. The advent of the digital economy has witnessed its increasingly pivotal role in economic development, with the total value of technology contract transactions serving as an indicator of the dynamism and developmental stage of the digital economy.

## 4. Discussions

Based on the established indicator system and analysis of rural revitalization, new-type urbanization, and the digital economy, the evolutionary trajectory of the coupling coordination degree across China's 31 provinces demonstrates a strong correlation with the respective levels of rural revitalization, new-type urbanization, and digital economy development within each province. The digital economy and urban-rural relations show a more coordinated trend over

time [56]. The digital economy has become the engine of rural revitalization development step by step, and the degree of coupling and coordination between the digital economy and rural revitalization is positively correlated with noticeable regional differences [6, 57, 58]. China's new-type urbanization and rural revitalization strategies have shifted from traditional territorial expansion to high-quality integration, with the degree of coupling and coordination between the two increasing and the quality of life in urban and rural areas consistent [59, 60]. These studies provide a theoretical basis for the coupled coordination degree relationship of rural revitalization, new-type urbanization, and digital economy, and when focusing on urban-rural development and digital economy development, policymakers and stakeholders must recognize the interactions between the three. As soon as possible, these studies have led to a deeper understanding of the relationship between urban and rural development and the digital economy. Still, the systemic integrity of the research has been neglected [33]. The current study focuses on exploring the relationship between the two two systems. Related research is mostly theoretical mechanism research; a few literature will be coupled coordination degree model applied to the relationship between the two, not the organic combination of these three, to explore the synergistic development relationship; in addition, the relevant research has not explored the obstacle factors of the three coupling, through the exploration of the obstacle factors, which can help to provide a precise basis for decision-making for the coordinated development of the digital economy, the new type of urbanization and the revitalization of the countryside.Still, the digital economy has become an essential engine of global economic change and a driving force for China's high-quality economic development.

At the same time, rural revitalization and new-type urbanization are fundamental initiatives to solve the three rural issues [61–64], so the coupled and coordinated relationship between the three is the focus of our inquiry. In this study, the relationship between rural revitalization, new-type urbanization, and digital economy is examined in depth from multiple perspectives, including the comprehensive development level, the spatio-temporal evolution of the coupling coordination degree, and the obstacle factors, with the aim of comprehensively and systematically evaluating and analyzing the coupling relationship between the digital economy and urban-rural development. Explore the obstacle factors for coupling the three, assist in high-quality economic development, green and sustainable development, and narrowing the gap between urban and rural areas in new urbanization and rural revitalization, and find opportunities for the innovative development of the digital economy.

When exploring the coupling of the three, it is found that the degree of coupling coordination develops from moderate dissonance to primary coordination over time and shows a spatial pattern of high in the east and low in the west. This is consistent with the results of the two-by-two system coupling, where the coupling coordination grades of rural revitalization and new-type urbanization [59], rural revitalization and digital economy [65], and digital economy and new-type urbanization evolve [29], and also reveal resource imbalances and strategic practices between different regional developments.

The level of technology, innovative talents, and innovative capital are key factors affecting new urbanization and rural revitalization; the number of postal workers and per capita food production are key factors affecting the digital economy and rural revitalization; the use of the Internet, mobile phones, personal computers, and telephones are strongly associated with urban-rural income disparity urban-rural development [33], and rural human capital is the key to affecting the digital economy and urban-rural integration [29]. When examining the obstacle factors affecting the coordination of the three couplings, it was found that the number of illiterate population/population aged 15 years and above, the index of the breadth of digital financial coverage, and the number of rural doctors and sanitary workers are the main obstacle factors, which also coincides with the main drivers of the coupling of the two-two systems.

## 5. Conclusions and policy suggestions

### 5.1 Conclusions

Dynamically grasping the coupled and coordinated relationship of rural revitalization, new-type urbanization, and digital economy systems is significant to developing Chinese villages and cities. Taking panel data from 31 provinces in China from 2011 to 2021 as a sample, this study measures the level of integrated rural revitalization-new urbanization-digital economy development, the degree of coupling, the degree of coupling coordination, and the analysis of obstacle factors, and investigates its spatio-temporal evolution characteristics. The main conclusions are as follows:

(1) The comprehensive evaluation level analysis shows that the comprehensive development level of the country's 31 provinces has been increasing yearly, but the overall level is low. In terms of subsystems, the comprehensive development level of rural revitalization has always been in the lead, rising from 0.3608 in 2011 to 0.4283 in 2021, and the digital economy ranks last; from 2011 to 2021, the highest is 0.1626, indicating that rural development has made great progress. Still, the development of the digital economy has not penetrated all aspects.

(2) An analysis of the temporal and spatial evolution of the coupling coordination degree shows that, in terms of time-series evolution characteristics, the degree of rural revitalization, new-type urbanization, and digital economy coupling, and the degree of coupling coordination show a fluctuating upward trend. The coupling degree increases from 0.6754 in 2011 to 0.8944 in 2021, and the coupling coordination degree increases from 0.0.3696 in 2011 to 0.5161 in 2021. Although the coupling coordination degree was dominated by moderate and mild disorder in 2011, it has developed into barely and primary coordination by 2021. Regarding spatial and temporal evolution, the degree of coordination shows a spatial evolution pattern of "high in the east and low in the west" and a concentration direction of "northeast-southwest," the development is becoming more and more balanced. The degree of coupling and coordination in the western region has increased, but the western region is affected by the resource environment, population quality, primary conditions, and others. The degree of coupling and coordination has always been in the lowest valley of the country.

(3) in terms of the main obstacle factors, the illiterate population/population aged 15 and above, the index of the breadth of digital financial coverage, and the number of rural doctors and sanitary workers are the main obstacle factors affecting the coupled and coordinated development of the three systems of rural revitalization, new-type urbanization, and digital economy. The per capita mileage of road routes and the total amount of technology contract transactions also have a more significant limiting effect on coordinating the coupling of the three systems.

### 5.2 Policy suggestions

Rural revitalization, new-type urbanization, and digital economy systems are essential means for people's well-being and solving urban-rural conflicts, as well as an important strategy for national development. Through the findings of this paper, countermeasure suggestions to promote the development of rural revitalization, new-type urbanization, and the digital economy are put forward.

First, promote the development of the digital economy. We need to take a series of scientific and reasonable measures to promote the sound development of China's digital economy and further enhance scientific research and innovation capacity. First, the government and enterprises should work closely together to formulate a long-term development plan for the digital economy, with clear development goals, critical tasks, and safeguard measures. On this basis,

we should increase investment in information infrastructure and enhance network coverage and transmission speed to provide a solid foundation for developing the digital economy. At the same time, we should actively promote the in-depth integration of digital technologies such as the Internet, big data, and artificial intelligence with traditional industries and take digital transformation as an opportunity to enhance industrial competitiveness and add value comprehensively. To stimulate innovation vitality, the government and enterprises should increase investment in scientific research and innovation, cultivate a high-level scientific research talent team, and promote the rapid transformation and application of scientific and technological achievements. Thus, developing the digital economy and enhancing scientific research and innovation are essential for China's economic and social development. Ensuring the sustained and robust development of China's digital economy requires ongoing reinforcement of policy guidance and strategic resource allocation. Moreover, fostering a collaborative ecosystem between government entities and private enterprises is crucial for cultivating an environment conducive to innovation and technological advancement, ultimately contributing significantly to the overall prosperity of China's economy and society.

Second, the government should formulate differentiated development to achieve coordinated regional development. Based on geographical advantages, resource advantages, and other factors, the coupling coordination degree of the eastern region is higher than that of the west, and the eastern region should continue to consolidate the achievements and give full play to the geographical advantages, resource advantages, and technical advantages; the degree of coupling coordination degree of the western region is lower, we should combine the regional characteristics, such as the development of characteristic rural tourism, camping tourism, etc., and integrate the digital economy to create a distinctive destination brand to promote rural revitalization and drive the new-type urbanization. To promote the coupled and coordinated development of rural revitalization, new-type urbanization, and digital economy, regions with a high degree of coupling coordination should fully drive the development of neighboring regions to bring strong and weak coordinated development.

Thirdly, addressing and mitigating the impediments to coupling coordination is imperative, thereby facilitating an enhancement of the overall level of coupling coordination. In the process of coupled and coordinated development of rural revitalization, new-type urbanization, and digital economy across the country, the number of illiterate population/population aged 15 years and above (y11), digital financial coverage breadth index (z7), and the number of rural doctors and sanitary workers (y8) are the main development obstacle factors. In order to promote the coupled and coordinated development level of the three, quality education should be actively promoted to improve the cultural level and ideology of the population. Secondly, the breadth of digital financial coverage should be improved. Finally, the number of rural doctors and health workers should be increased to improve the level of rural medical care.

This study mainly refers to the existing research results to construct a comprehensive evaluation index system of rural revitalization, new-type urbanization, and digital economy, and with the development of the countryside and cities, its indicators need to be constantly updated to establish better and more scientific evaluation indicators; this study takes the panel data of 31 provinces in China from 2011–2021 as the basis of the study, and the follow-up can be based on a single province as the research object, narrowing the scope of the study, focusing on the coupling and coordination degree of rural revitalization, new-type urbanization, and digital economy in each province, and more reasonably and accurately providing references for the integrated development of urban and rural areas in the digital era.

## Author Contributions

**Conceptualization:** Yajun Ma, Zhengyong Yu, Wei Liu.

**Data curation:** Qiang Ren.

**Formal analysis:** Zhengyong Yu, Wei Liu, Qiang Ren.

**Funding acquisition:** Zhengyong Yu.

**Investigation:** Qiang Ren.

**Methodology:** Yajun Ma, Zhengyong Yu.

**Software:** Wei Liu.

**Supervision:** Yajun Ma, Wei Liu, Qiang Ren.

**Writing – original draft:** Yajun Ma.

**Writing – review & editing:** Yajun Ma, Zhengyong Yu, Wei Liu.

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
