## [Decision Letter · Decision Letter 0]

2 Aug 2024

PONE-D-24-22029Exploring the coupling coordination relationship and obstacle factors of rural revitalization, new-type urbanization, and digital economy in ChinaPLOS ONE

Dear Dr. Liu,

Thank you for submitting your manuscript to PLOS ONE. After careful consideration, we feel that it has merit but does not fully meet PLOS ONE’s publication criteria as it currently stands. Therefore, we invite you to submit a revised version of the manuscript that addresses the points raised during the review process.

The rationality of indicator selection and the correlation between them need to be further identified, as the evaluation system created is the foundation of the entire study. I hope the author can carefully consider and provide corresponding modifications or reasonable explanations for this issue.Please further clarify the source of the data and provide a detailed description.The literature review should highlight the academic gap to fully discuss the similarities and differences with this study. In addition, the conclusion section needs further improvement in operability.Comprehensive language editing is suggested.==============================

We look forward to receiving your revised manuscript.

Kind regards,

Bifeng Zhu

Academic Editor

PLOS ONE

“The authors gratefully acknowledge the financial support funded by the Humanities and Social Science Fund of the Ministry of Education (Grant No. 22YJA630060, 22YJC790039) and supported by Yunnan Provincial Department of Education Science Research Fund (Grant No. 2024Y00504, 2023Y0363). This research is also supported by grants from Yunnan University's 15th Key Project of Scientific Research and Innovation (Grant No. KC-23235237).”

“The authors gratefully acknowledge the financial support funded by the Humanities and Social Science Fund of the Ministry of Education (Grant No. 22YJA630060, 22YJC790039) and supported by Yunnan Provincial Department of Education Science Research Fund (Grant No. 2024Y00504, 2023Y0363). This research is also supported by grants from Yunnan University's 15th Key Project of Scientific Research and Innovation (Grant No. KC-23235237).”

“The authors gratefully acknowledge the financial support funded by the Humanities and Social Science Fund of the Ministry of Education (Grant No. 22YJA630060, 22YJC790039) and supported by Yunnan Provincial Department of Education Science Research Fund (Grant No. 2024Y00504, 2023Y0363). This research is also supported by grants from Yunnan University's 15th Key Project of Scientific Research and Innovation (Grant No. KC-23235237).”

5. We note that Figures 5 and 6 in your submission contain [map/satellite] images which may be copyrighted. All PLOS content is published under the Creative Commons Attribution License (CC BY 4.0), which means that the manuscript, images, and Supporting Information files will be freely available online, and any third party is permitted to access, download, copy, distribute, and use these materials in any way, even commercially, with proper attribution. For these reasons, we cannot publish previously copyrighted maps or satellite images created using proprietary data, such as Google software (Google Maps, Street View, and Earth). For more information, see our copyright guidelines: http://journals.plos.org/plosone/s/licenses-and-copyright.

1. You may seek permission from the original copyright holder of Figures 5 and 6 to publish the content specifically under the CC BY 4.0 license. 

Reviewers' comments:

Reviewer's Responses to Questions

**Comments to the Author**

1. Is the manuscript technically sound, and do the data support the conclusions?

Reviewer #1: Partly

Reviewer #2: Yes

2. Has the statistical analysis been performed appropriately and rigorously? 

Reviewer #1: Yes

Reviewer #2: Yes

3. Have the authors made all data underlying the findings in their manuscript fully available?

Reviewer #1: Yes

Reviewer #2: No

4. Is the manuscript presented in an intelligible fashion and written in standard English?

Reviewer #1: No

Reviewer #2: Yes

5. Review Comments to the Author

Reviewer #1: 1、The subdivision indicators in the evaluation index system of the RR-NU-DE composite system established by the authors may have problems, mainly due to the correlation between indicators, not independence. The data of some subdivided indicators are not introduced how they are processed, for example, are the related indicators of GDP treated with price indices. The source of index data is also a simple introduction, not detailed, relatively general.

2、The author takes 31 provinces in China as samples for empirical research, but some index data of Tibet, Xinjiang, Qinghai and other provinces are missing for many years. The authors need to detail the data sources.

3、The discussion part and the suggestion part of the article are not detailed enough，the discussion section should involve the reasons for the results, and explain the similarities and differences with the conclusions of the existing literature, as well as the reasons for the differences. The recommendations given need to be tailored to each conclusion and suggest ways to improve the operability of the countermeasures. The author may need to mention more about the significance of the research for other countries and regions in the article, as well as the countermeasures and suggestions, and discuss the reference value for other countries and regions.

Reviewer #2: The structure of the article is reasonable, and the research method is appropriate. The authors give the data sources in this paper, but do not see the detailed data distribution. There are some grammatical errors and typos throughout the manuscript.

6. PLOS authors have the option to publish the peer review history of their article (what does this mean?). If published, this will include your full peer review and any attached files.

Reviewer #1: No

Reviewer #2: No

---

## [Author Response · Author response to Decision Letter 0]

6 Sep 2024

Dear reviewers,

Thank you very much for your careful review and detailed comments on our paper. The constructive suggestions helped us immensely in our revision. We have carefully considered these comments and answered all the questions one by one. We revised the manuscript accordingly. The main revised part is highlighted in red in the manuscript. 

Thank you and best regards!

Yours Sincerely,

Authors

Reviewer #1

● 1- - (1) The subdivision indicators in the evaluation index system of the RR-NU-DE composite system established by the authors may have problems, mainly due to the correlation between indicators, not independence. The data of some subdivided indicators are not introduced how they are processed, for example, are the related indicators of GDP treated with price indices. The source of index data is also a simple introduction, not detailed, relatively general.

Response : Thank you for your comments. We have made detailed changes to the problem, and revised the manuscript. In order to more clearly understand the correlation between the evaluation index system, this paper adds Pearson correlation analysis. By measuring the Pearson correlation coefficient between the indicators, the correlation between the indicators is expressed. Through Pearson correlation analysis, the correlation heat map of the three systems of rural revitalization, new urbanization and digital economy is obtained.

Figure 2 Correlation analysis of Pearson coefficient between the characteristics of rural revitalization dimension

Fig.3 Pearson coefficient correlation analysis between the characteristics of the new urbanization dimension

 Figure 4 Pearson coefficient correlation analysis between the characteristics of the digital economy dimension

From Figure 2 and Figure 4, it can be seen that the correlation coefficients of each index in the rural revitalization system and the new urbanization system are concentrated between [ 0,1 ], with obvious correlation. In Figure 3, the correlation coefficients of each index in the digital economy system are all above 0, and all indexes are correlated.

● 2- - (2) The author takes 31 provinces in China as samples for empirical research, but some index data of Tibet, Xinjiang, Qinghai and other provinces are missing for many years. The authors need to detail the data sources.

Response : Thank you for your comments. Due to historical and locational factors, relevant data are missing in some regions, such as Xinjiang and Tibet, and this study uses the interpolation method to fill in and make up for the shortcomings of the simple average method.

● 3- - (3)The discussion part and the suggestion part of the article are not detailed enough，the discussion section should involve the reasons for the results, and explain the similarities and differences with the conclusions of the existing literature, as well as the reasons for the differences. The recommendations given need to be tailored to each conclusion and suggest ways to improve the operability of the countermeasures. The author may need to mention more about the significance of the research for other countries and regions in the article, as well as the countermeasures and suggestions, and discuss the reference value for other countries and regions.

Response: Thank you for your comments. In view of the lack of detailed discussion and suggestions, we have carefully revised it and compared the problems with the conclusions and countermeasures. Through discussion and suggestions, it puts forward feasible reference value for the development of other countries.

Modified paragraph

According to the indicator system and analysis results of rural revitalization, new-type urbanization, and digital economy, the evolution of the coupling coordination degree level of the 31 provinces in the country is closely related to the rural revitalization, new-type urbanization, and digital economy of each province. The digital economy and urban-rural relations show a more coordinated trend over time 75. The digital economy has become the engine of rural revitalization development step by step, and the degree of coupling and coordination between the digital economy and rural revitalization is positively correlated with noticeable regional differences 7,81,82. China's new-type urbanization and rural revitalization strategies have shifted from traditional territorial expansion to high-quality integration, with the degree of coupling and coordination between the two increasing and the quality of life in urban and rural areas consistent 83,84. These studies provide a theoretical basis for the coupled coordination degree relationship of rural revitalization, new-type urbanization, and digital economy, and when focusing on urban-rural development and digital economy development, policymakers and stakeholders must recognize the interactions between the three. As soon as possible these studies have led to a deeper understanding of the relationship between urban and rural development and the digital economy. Still, the systemic integrity of the research has been neglected 44. The current study focuses on exploring the relationship between the two two systems, but the digital economy has become an essential engine of global economic change and a driving force for China's high-quality economic development, while rural revitalization and new-type urbanization are fundamental initiatives to solve the three rural issues 2,85-87, so the coupled and coordinated relationship between the three is the focus of our inquiry. In this study, the relationship between rural revitalization, new-type urbanization, and digital economy is examined in depth from multiple perspectives, including the comprehensive development level, the spatio-temporal evolution of the coupling coordination degree, and the obstacle factors, with the aim of comprehensively and systematically evaluating and analyzing the coupling relationship between the digital economy and urban-rural development.Explore the obstacle factors for the coupling of the three, provide assistance for the high-quality economic development, green and sustainable development, and narrowing the gap between urban and rural areas in new urbanization and rural revitalization, and find opportunities for the innovative development of the digital economy.

When exploring the coupling of the three, it is found that the degree of coupling coordination develops from moderate dissonance to primary coordination over time and shows a spatial pattern of high in the east and low in the west. This is consistent with the results of the two-by-two system coupling, where the coupling coordination grades of rural revitalization and new-type urbanization 83, rural revitalization and digital economy 88, and digital economy and new-type urbanization evolve 37, and also reveal resource imbalances and strategic practices between different regional developments.

The level of technology, innovative talents and innovative capital are key factors affecting new urbanization and rural revitalization; the number of postal workers and per capita food production are key factors affecting the digital economy and rural revitalization; the use of the Internet, mobile phones, personal computers and telephones are strongly associated with urban-rural income disparity urban-rural development 44, and rural human capital is the key to affecting the digital economy and urban-rural integration 37. When examining the obstacle factors affecting the coordination of the three couplings, it was found that the number of illiterate population/population aged 15 years and above, the index of the breadth of digital financial coverage, and the number of rural doctors and sanitary workers are the main obstacle factors, which also coincides with the main drivers of the coupling of the two-two systems.

Dynamically grasping the coupled and coordinated relationship of rural revitalization, new-type urbanization, and digital economy systems is of great significance to the development of Chinese villages and cities. Taking panel data from 31 provinces in China from 2011 to 2021 as a sample, this study measures the level of integrated rural revitalization-new urbanization-digital economy development, the degree of coupling, the degree of coupling coordination, and the analysis of obstacle factors, and investigates its spatio-temporal evolution characteristics. The main conclusions are as follows:

(1) Through the comprehensive evaluation level analysis, it can be seen that he comprehensive development level of the country's 31 provinces has been increasing yearly, but the overall level is low. In terms of subsystems, the comprehensive development level of rural revitalization has always been in the lead, rose from 0.3608 in 2011 to 0.4283 in 2021,and the digital economy ranks last, From 2011 to 2021, the highest is 0.1626,indicating that rural development has made great progress. Still, the development of the digital economy has not penetrated all aspects.

(2) An analysis of the temporal and spatial evolution of the coupling coordination degree shows that,in terms of time-series evolution characteristics, the degree of rural revitalization, new-type urbanization, and digital economy coupling and the degree of coupling coordination show a fluctuating upward trend. The coupling degree increases from 0.6754 in 2011 to 0.8944 in 2021, and the coupling coordination degree increases from 0.0.3696 in 2011 to 0.5161 in 2021.Although the coupling coordination degree was dominated by moderate and mild disorder in 2011, it has developed into barely and primary coordination by 2021. in terms of spatial and temporal evolution, the degree of coordination shows a spatial evolution pattern of "high in the east and low in the west" and a concentration direction of "northeast-southwest," and the development is becoming more and more balanced. The degree of coupling and coordination in the western region has increased, but the western region is affected by the resource environment, population quality, primary conditions, and other conditions. The degree of coupling and coordination has always been in the lowest valley of the country.

(3) in terms of the main obstacle factors, the illiterate population/population aged 15 and above, the index of the breadth of digital financial coverage, and the number of rural doctors and sanitary workers are the main obstacle factors affecting the coupled and coordinated development of the three systems of rural revitalization, new-type urbanization, and digital economy. The per capita mileage of road routes and the total amount of technology contract transactions also have a more significant limiting effect on coordinating the coupling of the three systems.

Rural revitalization, new-type urbanization, and digital economy systems are essential means for people's well-being and solving urban-rural conflicts, as well as an important strategy for national development. Through the findings of this paper, countermeasure suggestions to promote the development of rural revitalization, new-type urbanization, and the digital economy are put forward.

First, promote the development of the digital economy. We need to take a series of scientific and reasonable measures to promote the sound development of China's digital economy and further enhance scientific research and innovation capacity. First, the government and enterprises should work closely together to formulate a long-term development plan for the digital economy, with clear development goals, critical tasks, and safeguard measures. On this basis, we should increase investment in information infrastructure and enhance network coverage and transmission speed to provide a solid foundation for developing the digital economy. At the same time, we should actively promote the in-depth integration of digital technologies such as the Internet, big data, and artificial intelligence with traditional industries and take digital transformation as an opportunity to enhance industrial competitiveness and add value comprehensively. To stimulate innovation vitality, the government and enterprises should increase investment in scientific research and innovation, cultivate a high-level scientific research talent team, and promote the rapid transformation and application of scientific and technological achievements. In conclusion, developing the digital economy and enhancing scientific research and innovation are important tasks facing China's economic and social development. Only by continuously strengthening policy guidance and resource investment can we support the sustained and healthy development of China's digital economy. The government and enterprises should work together to create a more favorable environment for innovation and development and make a greater contribution to the prosperity of China's economy and society.

Second, the government should formulate differentiated development to achieve coordinated regional development. Based on geographical advantages, resource advantages, and other factors, the coupling coordination degree of the eastern region is higher than that of the west, and the eastern region should continue to consolidate the achievements and give full play to the geographical advantages, resource advantages, and technical advantages; the degree of coupling coordination degree of the western region is lower, we should combine the regional characteristics, such as the development of characteristic rural tourism, camping tourism, etc., and integrate the digital economy to create a distinctive destination brand to promote rural revitalization and drive the new-type urbanization. In order to promote the coupled and coordinated development of rural revitalization, new-type urbanization, and digital economy, regions with a high degree of coupling coordination should fully drive the development of neighboring regions to bring strong and weak coordinated development.

Thirdly, the obstacles to coupling coordination should be unblocked, and the degree of coupling coordination should be improved. In the process of coupled and coordinated development of rural revitalization, new-type urbanization, and digital economy across the country, the number of illiterate population/population aged 15 years and above (y11), digital financial coverage breadth index (z7), and the number of rural doctors and sanitary workers (y8) are the main development obstacle factors. In order to promote the coupled and coordinated development level of the three, quality education should be actively promoted to improve the cultural level and ideology of the population. Secondly, the breadth of digital financial coverage should be improved. Finally, the number of rural doctors and health workers should be increased to improve the level of rural medical care.

This study mainly refers to the existing research results to construct a comprehensive evaluation index system of rural revitalization, new-type urbanization, and digital economy, and with the development of the countryside and cities, its indicators need to be constantly updated to establish better and more scientific evaluation indicators; this study takes the panel data of 31 provinces in China from 2011-2021 as the basis of the study, and the follow-up can be based on a single province as the research object, narrowing the scope of the study, focusing on the coupling and coordination degree of rural revitalization, new-type urbanization, and digital economy in each province, and more reasonably and accurately providing references for the integrated development of urban and rural areas in the digital era.

Reviewer #2: 1.The structure of the article is reasonable, and the research method is appropriate. The authors give the data sources in this paper, but do not see the detailed data distribution. There are some grammatical errors and typos throughout the manuscript.

Response : Thank you for your comments. The article has been checked and modified in detail for syntax and comment errors.

2.There are many problems with the use of English. For example, there are numerous grammatical and typographical errors scattered throughout the manuscript (e.g., composite is used inappropriately and should be replaced with comprehensive, “comprehe

---

## [Decision Letter · Decision Letter 1]

9 Oct 2024

PONE-D-24-22029R1Exploring the coupling coordination relationship and obstacle factors of rural revitalization, new-type urbanization, and digital economy in ChinaPLOS ONE

Dear Dr. Liu,

Thank you for submitting your manuscript to PLOS ONE. After careful consideration, we feel that it has merit but does not fully meet PLOS ONE’s publication criteria as it currently stands. Therefore, we invite you to submit a revised version of the manuscript that addresses the points raised during the review process.

Unfortunately, the author's revisions did not satisfy the reviewer. Please make in-depth revisions to the reviewer's comments. This will also be the last opportunity for a comprehensive revision.==============================

We look forward to receiving your revised manuscript.

Kind regards,

Bifeng Zhu

Academic Editor

PLOS ONE

Journal Requirements:

Reviewers' comments:

Reviewer's Responses to Questions

**Comments to the Author**

1. If the authors have adequately addressed your comments raised in a previous round of review and you feel that this manuscript is now acceptable for publication, you may indicate that here to bypass the “Comments to the Author” section, enter your conflict of interest statement in the “Confidential to Editor” section, and submit your "Accept" recommendation.

Reviewer #1: (No Response)

2. Is the manuscript technically sound, and do the data support the conclusions?

Reviewer #1: (No Response)

3. Has the statistical analysis been performed appropriately and rigorously? 

Reviewer #1: (No Response)

4. Have the authors made all data underlying the findings in their manuscript fully available?

Reviewer #1: (No Response)

5. Is the manuscript presented in an intelligible fashion and written in standard English?

Reviewer #1: (No Response)

6. Review Comments to the Author

Reviewer #1: 1. There are still some problems in the article, for example, the introduction needs to introduce the research significance of the article from a global perspective.

2. The discussion part of the paper needs to be further deepened. What is the difference between the conclusions of this paper and those of existing research? It is necessary to analyze the reasons for the difference.

3. The language of the article still needs polishing.

4. The vast majority of references are articles by Chinese scholars, and the small number of articles by scholars from other countries indicates that scholars from other countries are not interested in the topic?

7. PLOS authors have the option to publish the peer review history of their article (what does this mean?). If published, this will include your full peer review and any attached files.

Reviewer #1: No

---

## [Author Response · Author response to Decision Letter 1]

15 Oct 2024

Dear reviewers,

Thank you very much for your careful review and detailed comments on our paper. The constructive suggestions helped us immensely in our revision. We have carefully considered these comments and answered all the questions one by one. We revised the manuscript accordingly. The main revised part is highlighted in red in the manuscript. 

Thank you and best regards!

Yours Sincerely,

Authors

Reviewer #1

● 1- - (1) There are still some problems in the article, for example, the introduction needs to introduce the research significance of the article from a global perspective.

Response: Thank you for your comments. The introductory part of the article adds the significance of exploring the coupling relationship between the three systems of rural revitalization-new urbanization-digital economy from a global perspective

Modified Introduction:

Inequality is a global concern of the United Nations Sustainable Development Goals, and the digital economy (DE) is helping to reduce a country’s income gap on a global scale 1. With the arrival of the digital economy, the number of global Internet users has reached 5.373 billion. The digital economy breaks the limitations of traditional modes, weakens the asymmetry of information access, and becomes an efficient driving force for economic efficiency and quality improvement 2. The United Nations introduced the Sustainable Development Goals (SDGs), which, in order to accelerate progress and achievements, specifically emphasize the importance of the countryside and promote rural revitalization (RR) as a critical strategic objective 3,4. The ultimate goal of the rural revitalization strategy is common prosperity, which solves the problem of unbalanced development between urban and rural areas. New-type urbanization (NU), as an exogenous power of the countryside, promotes the flow of talents, science and technology, capital, information, and other factors to the countryside, stimulates the endogenous power of the countryside, and realizes the revitalization of the countryside. RR is the foundation and premise of new urbanization, which brings new space carriers and demand power for the development of DE. At the same time, DE provides new kinetic energy, such as information and resources for the development of rural revitalization. In the process of promoting rural revitalization, accelerating new urbanization and developing digital economy, exploring the coupled and coordinated relationship, spatial and temporal evolution pattern, and the obstacle factors of the coordinated development in the RR-NU-DE system will help to accelerate the realization of a series of goals, such as China’s economic transformation, digitalization, urban-rural integration, and rural revitalization, and at the same time refer to the digital economy and urban-rural integration in the world. From a global perspective, many countries are currently facing unbalanced development between urban and rural areas, aging populations, technological backwardness and other dilemmas, and exploring the coupling relationship between rural revitalization, new urbanization and the digital economy, as well as the main obstacles, can effectively reduce the gap between urban and rural areas, improve the overall competitiveness of the country, and promote the flow of resources between regions, thereby promoting the balanced development of all countries globally and realizing the optimization of the allocation of resources globally.

● 2- - (2) The discussion part of the paper needs to be further deepened. What is the difference between the conclusions of this paper and those of existing research? It is necessary to analyze the reasons for the difference.

Response: Thank you for your comments. The article has been refined with a conclusion section that highlights the importance of this study by contrasting it with existing findings.

Modified Discussions:

According to the indicator system and analysis results of rural revitalization, new-type urbanization, and digital economy, the evolution of the coupling coordination degree level of the 31 provinces in the country is closely related to the rural revitalization, new-type urbanization, and digital economy of each province. The digital economy and urban-rural relations show a more coordinated trend over time 77. The digital economy has become the engine of rural revitalization development step by step, and the degree of coupling and coordination between the digital economy and rural revitalization is positively correlated with noticeable regional differences 7,83,84. China’s new-type urbanization and rural revitalization strategies have shifted from traditional territorial expansion to high-quality integration, with the degree of coupling and coordination between the two increasing and the quality of life in urban and rural areas consistent 85,86. These studies provide a theoretical basis for the coupled coordination degree relationship of rural revitalization, new-type urbanization, and digital economy, and when focusing on urban-rural development and digital economy development, policymakers and stakeholders must recognize the interactions between the three. As soon as possible, these studies have led to a deeper understanding of the relationship between urban and rural development and the digital economy. Still, the systemic integrity of the research has been neglected 46. The current study focuses on exploring the relationship between the two two systems. Related research is mostly theoretical mechanism research, a few literature will be coupled coordination degree model applied to the relationship between the two, not the organic combination of these three, to explore the synergistic development relationship; in addition, the relevant research has not explored the obstacle factors of the three coupling, through the exploration of the obstacle factors, which can help to provide a precise basis for decision-making for the coordinated development of the digital economy, the new type of urbanization and the revitalization of the countryside. Still, the digital economy has become an essential engine of global economic change and a driving force for China’s high-quality economic development. At the same time, rural revitalization and new-type urbanization are fundamental initiatives to solve the three rural issues 2,87-89, so the coupled and coordinated relationship between the three is the focus of our inquiry. In this study, the relationship between rural revitalization, new-type urbanization, and digital economy is examined in depth from multiple perspectives, including the comprehensive development level, the spatio-temporal evolution of the coupling coordination degree, and the obstacle factors, with the aim of comprehensively and systematically evaluating and analyzing the coupling relationship between the digital economy and urban-rural development. Explore the obstacle factors for coupling the three, assist in high-quality economic development, green and sustainable development, and narrowing the gap between urban and rural areas in new urbanization and rural revitalization, and find opportunities for the innovative development of the digital economy.

● 3- - (3) The language of the article still needs polishing.

Response: Thank you for your comments. We have carefully checked the language of the full manuscript. And we also have revised the abstract. According to your advice, this manuscript was edited for proper English language, grammar, punctuation, spelling, and overall style by one or more highly qualified native English speakers.

● 4- - (4) The vast majority of references are articles by Chinese scholars, and the small number of articles by scholars from other countries indicates that scholars from other countries are not interested in the topic?

Response: Thank you for your insightful comments regarding the references cited in our manuscript. We appreciate your observation about the predominance of articles by Chinese scholars and the implications it may have on the perceived interest of international researchers in this topic. In response, we will expand our literature review to include a broader range of studies from scholars outside of China. This will provide a more comprehensive overview of global perspectives on the subject and demonstrate the relevance and importance of our research within the international academic community. We believe that incorporating a wider array of references will strengthen our work and reflect the global significance of our study. Thank you once again for your valuable feedback.

---

## [Decision Letter · Decision Letter 2]

21 Oct 2024

Exploring the coupling coordination relationship and obstacle factors of rural revitalization, new-type urbanization, and digital economy in China

PONE-D-24-22029R2

Dear Dr. Liu,

We’re pleased to inform you that your manuscript has been judged scientifically suitable for publication and will be formally accepted for publication once it meets all outstanding technical requirements.

Kind regards,

Bifeng Zhu

Academic Editor

PLOS ONE

Additional Editor Comments (optional):

Reviewers' comments:

Reviewer's Responses to Questions

**Comments to the Author**

1. If the authors have adequately addressed your comments raised in a previous round of review and you feel that this manuscript is now acceptable for publication, you may indicate that here to bypass the “Comments to the Author” section, enter your conflict of interest statement in the “Confidential to Editor” section, and submit your "Accept" recommendation.

Reviewer #1: (No Response)

2. Is the manuscript technically sound, and do the data support the conclusions?

Reviewer #1: (No Response)

3. Has the statistical analysis been performed appropriately and rigorously? 

Reviewer #1: (No Response)

4. Have the authors made all data underlying the findings in their manuscript fully available?

Reviewer #1: (No Response)

5. Is the manuscript presented in an intelligible fashion and written in standard English?

Reviewer #1: (No Response)

6. Review Comments to the Author

Reviewer #1: (No Response)

7. PLOS authors have the option to publish the peer review history of their article (what does this mean?). If published, this will include your full peer review and any attached files.

Reviewer #1: No

---

## [Editor Report · Acceptance letter]

28 Oct 2024

PONE-D-24-22029R2 

PLOS ONE

Dear Dr. Liu, 

I'm pleased to inform you that your manuscript has been deemed suitable for publication in PLOS ONE. Congratulations! Your manuscript is now being handed over to our production team.

Kind regards, 

on behalf of

Dr. Bifeng Zhu 

Academic Editor

PLOS ONE